# StethoLM: Audio Language Model for Cardiopulmonary Analysis Across Clinical Tasks

**Yishan Wang**                                                     *y.wang18@tue.nl*
*Eindhoven University of Technology*

**Tsai-Ning Wang**                                                  *t.n.wang@tue.nl*
*Eindhoven University of Technology*

**Mathias Funk**                                                    *m.funk@tue.nl*
*Eindhoven University of Technology*

**Aaqib Saeed**                                                     *a.saeed@tue.nl*
*Eindhoven University of Technology*

**Reviewed on OpenReview:** *https://openreview.net/forum?id=i9RuUH9Jyj*

## Abstract

Listening to heart and lung sounds — *auscultation* — is one of the first and most fundamental steps in a clinical examination. Despite being fast and non-invasive, it demands years of experience to interpret subtle audio cues. Recent deep learning methods have made progress in automating cardiopulmonary sound analysis, yet most are restricted to simple classification and offer little clinical interpretability or decision support. We present StethoLM, the first audio–language model specialized for cardiopulmonary auscultation, capable of performing instruction-driven clinical tasks across the full spectrum of auscultation analysis. StethoLM integrates audio encoding with a medical language model backbone and is trained on StethoBench, a comprehensive benchmark comprising 77,027 instruction–response pairs synthesized from 16,125 labeled cardiopulmonary recordings spanning seven clinical task categories: binary classification, detection, reporting, reasoning, differential diagnosis, comparison, and location-based analysis. Through multi-stage training that combines supervised fine-tuning and direct preference optimization, StethoLM achieves substantial gains in performance and robustness on out-of-distribution data. Our work establishes a foundation for instruction-following AI systems in clinical auscultation.

## 1 Introduction

Auscultation, the practice of listening to heart and lung sounds with a stethoscope, has long been a cornerstone of clinical examination. It enables physicians to detect murmurs, wheezes, crackles, and other acoustic markers that signal cardiopulmonary disease. Unlike imaging or laboratory testing, auscultation data can be obtained rapidly, non-invasively, and at low cost, making it a uniquely scalable modality for screening, triage, and longitudinal monitoring, especially in low-resource settings. Yet, interpreting these sounds requires significant clinical expertise, with physicians needing years of training to reliably identify subtle abnormalities and develop sound clinical judgment. This expertise barrier limits the scalability of auscultation: many healthcare settings lack access to experienced clinicians, training is time-intensive, and diagnostic consistency varies with experience level. These human resource constraints motivate the development of AI systems that could augment clinical expertise or extend auscultation capabilities to settings with limited specialist access.

To address this challenge, contemporary deep learning approaches have shown that automated analysis of cardiopulmonary sounds is feasible (Ma et al., 2022; Acharya & Basu, 2020a). However, the vast majority

of research has been constrained by a *classification paradigm*: models are trained to perform narrow tasks (e.g., "murmur vs. no murmur") and output a single, fixed label (Zhang et al., 2024b). This paradigm fundamentally misaligns with clinical practice. The task of a clinician is to describe acoustic findings, compare recordings over time, differentiate between similar-sounding pathologies, and reason about underlying causes. Models confined to single-label classification cannot support this diverse range of clinical reasoning tasks, which has limited their adoption in clinical settings (Labkoff et al., 2024).

Recent advances in multimodal large language models (Xu et al., 2025a) offer an opportunity to break free from this restrictive paradigm. By combining auditory perception with language-based reasoning, these models can, in principle, perform instruction-driven tasks and generate interpretable, free-text outputs that emulate clinical dialogue (Wang et al., 2023). However, while general-purpose models can process generic audio, they lack the specialized knowledge required for auscultation. Medical audio presents unique challenges distinct from music, speech, or environmental sounds: diagnostically relevant features reside in millisecond-scale temporal patterns (fine crackles <5ms vs. coarse crackles >10ms), subtle spectral characteristics (monophonic vs. polyphonic wheezing), and phase-specific timing (systolic vs. diastolic murmurs) (Kim et al., 2021; Ma et al., 2022). These fine-grained acoustic distinctions, absent from general audio tasks, require specialized training on medical data. Existing medical multimodal models have focused primarily on vision-language tasks or limited audio Q&A (Wang et al., 2025; Zhang et al., 2024b), leaving a critical gap: no system is capable of performing the diverse clinical reasoning tasks required for expert-level cardiopulmonary auscultation.

Here, we introduce a new approach for AI in auscultation, shifting the focus from classification to comprehensive clinical reasoning. To this end, we present StethoLM (see Figure 1 for an overview), the first audio-language model specialized for cardiopulmonary analysis, and StethoBench, a benchmark designed to train and evaluate models on multiple clinical reasoning tasks within cardiopulmonary analysis. Our key contributions are:

- **StethoLM**, an audio-language model designed specifically for cardiopulmonary auscultation (see Figure 1), capable of performing seven distinct, instruction-driven clinical reasoning tasks, from detection and reporting to differential diagnosis and comparison[1][2].

- **StethoBench**, a new, comprehensive benchmark of 77,027 instruction-response pairs derived from 16,125 clinical recordings. It is the first benchmark designed to move beyond classification and facilitate the development of generalist auscultation models[3].

- **Extensive experiments** showing that specialized, instruction-based training is critical. StethoLM substantially outperforms powerful, general-purpose multimodal models on both in-domain and out-of-domain data, establishing a new state-of-the-art for multi-task clinical audio understanding.

## 2 Related work

**Machine Learning for Cardiopulmonary Auscultation** Machine learning has been extensively applied to cardiopulmonary sounds, but historically, this research has been dominated by a classification paradigm. Early efforts used signal processing and shallow classifiers to distinguish between categories such as "normal vs. abnormal" heart sounds or "wheeze vs. crackle" in lung recordings (Ma et al., 2022). With the advent of deep learning, models trained on spectrogram representations achieved strong performance on public benchmarks, including the PhysioNet/CinC Challenge for heart sound classification and the ICBHI Challenge for respiratory sound analysis (Li et al., 2022). More recent work has leveraged transfer learning and self-supervised audio representations to compensate for small dataset sizes, further improving accuracy on these tasks (Gairola et al., 2021; Acharya & Basu, 2020a). Current approaches remain fragmented and label-centric, optimized only for narrow diagnostic categories, thus provide little support for broader clinical decision-making.

---

[1]Code: https://github.com/yishani/StethoLM
[2]Model: https://huggingface.co/askyishan/StethoLM
[3]Dataset: https://huggingface.co/datasets/askyishan/StethoBench

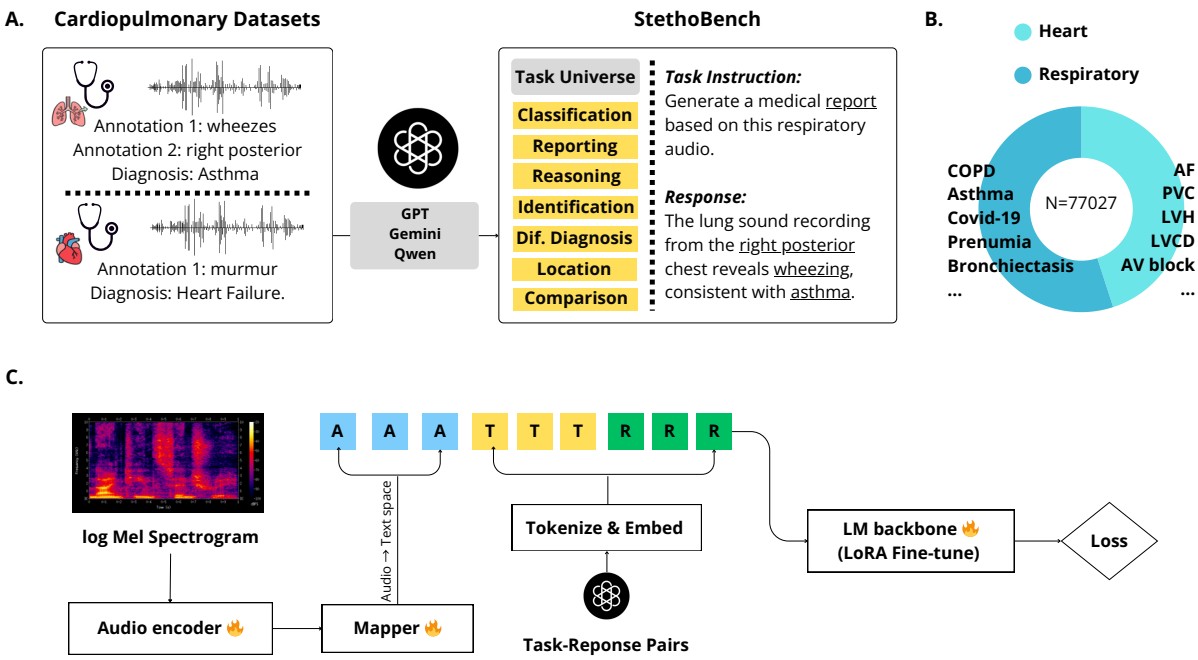

Figure 1: **Overview of StethoLM and StethoBench.** A. Automated benchmark creation pipeline, where off-the-shelf LLMs generate 77,027 task–response pairs from 16,125 cardiopulmonary recordings and associated annotations. B. Distribution of audio type and the examples of disease that StethoLM covers. C. StethoLM architecture integrating audio–text alignment, supervised fine-tuning for diverse clinical tasks.

**General-Purpose Audio-Language Models** Foundation audio-language models have recently enabled reasoning over complex auditory inputs. Models such as Pengi (Deshmukh et al., 2023) use audio embeddings as conditioning tokens to perform audio-to-text tasks, while LTU (Gong et al., 2023) and more recent systems like GAMA (Ghosh et al., 2024) and Qwen2.5-Omni (Xu et al., 2025a) leverage pre-trained audio encoders to integrate auditory signals with language models, enabling tasks where the model follows natural-language instructions about audio. Yet these models are primarily trained on general-purpose audio, speech, or environmental sounds. They are not designed for domain-specific tasks such as cardiopulmonary auscultation, where fine-grained acoustic patterns, and clinical metadata are critical. As a result, their direct applicability to medical audio is limited, motivating the need for domain-specific audio-language modeling.

**Multimodal Large Language Models for Healthcare** Instruction-tuned large language models in healthcare, such as MedPaLM (Singhal et al., 2023) and ClinicalGPT (Wang et al., 2023), demonstrate that LLMs can integrate domain knowledge and generate structured outputs for complex medical tasks. More recent multimodal systems, including MedGemma (Sellergren et al., 2025) and MedRAX (Fallahpour et al., 2025), extend these capabilities to visual–language inputs, supporting applications such as diagnosis, report generation, and multi-step clinical reasoning. Together, these works illustrate the potential of leveraging LLMs to move beyond fragmented, label-centric outputs toward multitask clinical support. In the cardiopulmonary domain, models such as RespLLM (Zhang et al., 2024b) and preliminary Q&A systems have attempted to adapt LLMs for respiratory and cardiac audio (Wang et al., 2025), but they remain limited in task diversity and scope, underscoring the need for a more comprehensive, generalist approach.

**Clinical Tasks in Medical AI: From Specialized to Generalist Systems** In clinical practice, auscultation involves multiple interconnected tasks. Physicians must assess whether the sounds are normal or abnormal, localize specific events (e.g., "S1 at mitral area," "crackles during inspiration"), summarize findings in clinical notes, reason about underlying pathology, and compare recordings across visits. Existing work

has trained medical AI models for these specialized tasks. Classification and detection models for cardiopulmonary sounds achieve strong performance on specific sound identification (e.g., wheeze vs. crackle) (Kim et al., 2021; Acharya & Basu, 2020b). Report generation—translating findings into structured text—has been explored in radiology (Wang et al., 2024; Ouis & Akhloufi, 2024) but is not yet available for clinical audio. Differential diagnosis systems generate ranked diagnosis lists from clinical data (McDuff et al., 2025), while explainable AI methods provide reasoning for model predictions (Sadeghi et al., 2024)—both critical for clinical trust but typically isolated from other tasks. Longitudinal and comparative analysis, essential for tracking disease progression, has been studied primarily for EHR data (Cascarano et al., 2023). These capabilities reflect distinct aspects of routine clinical work, yet they are rarely integrated. In cardiopulmonary auscultation, this fragmentation is particularly obvious: most systems address only single-label classification, leaving broader clinical tasks and workflows unsupported.

## 3 Problem Formulation

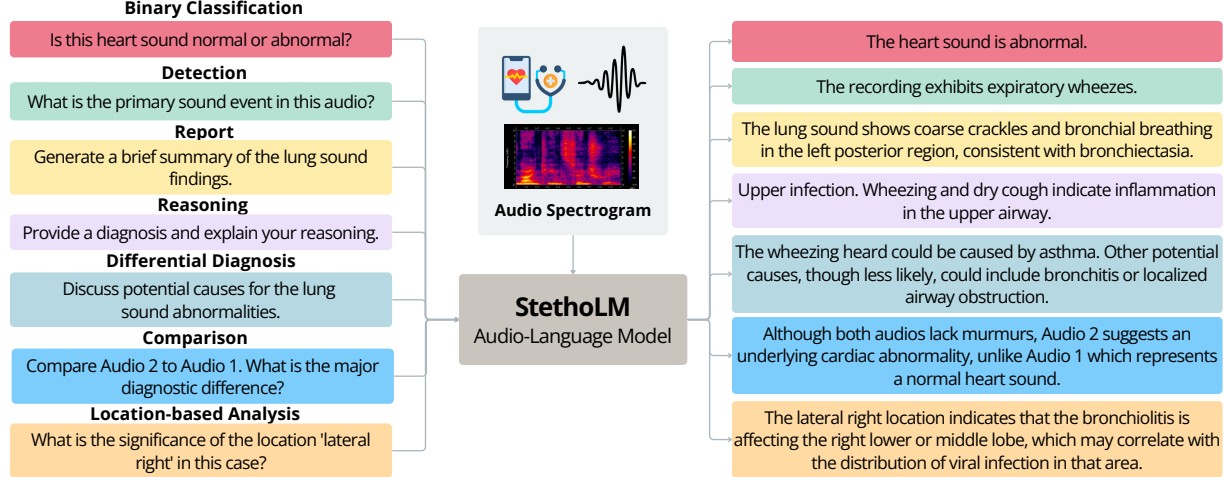

Figure 2: **Diverse clinical tasks supported by StethoLM.** Instructions (left) represent realistic clinical queries, while responses (right) provide task-appropriate outputs ranging from binary decisions to complex diagnostic reasoning.

We formulate cardiopulmonary auscultation as a conditional language modeling problem. Given an audio recording $\mathbf{x} \in \mathbb{R}^T$ and a natural language instruction $\mathbf{q} = (q_1, \ldots, q_K)$ consisting of $K$ tokens, the goal is to generate a textual response $\mathbf{y} = (y_1, \ldots, y_L)$ of $L$ tokens that accurately addresses the instruction. Our model, parameterized by $\theta$, learns the conditional probability distribution $p_\theta(\mathbf{y}|\mathbf{x}, \mathbf{q})$.

Due to the sequential nature of language, this distribution is decomposed autoregressively using the chain rule of probability:

$$p_\theta(\mathbf{y}|\mathbf{x}, \mathbf{q}) = \prod_{j=1}^{L} p_\theta(y_j|\mathbf{y}_{<j}, \mathbf{x}, \mathbf{q}), \tag{1}$$

where $\mathbf{y}_{<j} = (y_1, \ldots, y_{j-1})$ are the previously generated tokens. This formulation differs fundamentally from traditional label-centric approaches, which learn a fixed mapping $\mathbf{x} \to c \in \mathcal{C}$ for a predefined set of classes $\mathcal{C}$. Our approach instead enables the model to interpret diverse instructions $\mathbf{q}$ and generate appropriate free-form responses.

The model is trained on $\mathcal{D} = \{(\mathbf{x}_i, \mathbf{q}_i, \mathbf{y}_i)\}_{i=1}^{N}$, where each triplet contains an audio sample, an instruction, and the corresponding ground-truth response. Our objective is to maximize the likelihood of generating

Table 1: **Summary of datasets included in StethoBench.** The upper section lists in-distribution (ID) datasets used for training, validation, and testing. The lower section lists out-of-distribution (OOD) datasets used for generalization evaluation. OOD datasets differ from ID data in recording devices, patient populations, disease prevalence, clinical contexts, or collection protocols, reflecting realistic deployment scenarios. "Samples" indicates the number of *audio recordings.* "Pairs" refers to the number of corresponding *instruction-response pairs.*

| Dataset | Samples (train/val/test) | Pairs | Minutes | Modality | Application / Task |
|---------|--------------------------|-------|---------|----------|--------------------|
| *In-Distribution (ID) Datasets* | | | | | |
| ICBHI | 539 / 381 | 9,028 | 329 | Respiratory | Abnormality Detection |
| KAUH | 200 / 69 / 67 | 5,842 | 97 | Mixed | Disease Classification |
| SPR | 1,773 / 1,002 / 48 | 14,183 | 276 | Respiratory | Respiratory Event Detection |
| CoughVID | 1,223 / 830 | 9,844 | 283 | Cough | COVID-19 Detection |
| CovidUK | 2,000 / 1,000 / 1,000 | 13,356 | 378 | Cough | COVID-19 Screening |
| ZCH | 1,019 / 240 | 6,387 | 258 | Cardiac | Heart Sound Analysis |
| CirCor | 1,560 / 1,447 | 14,466 | 1,140 | Cardiac | Murmur Detection |
| *Out-of-Distribution (OOD) Datasets* | | | | | |
| TR | 420 | 856 | 141 | Respiratory | COPD Severity Assessment |
| CinC | 442 | 900 | 174 | Cardiac | Abnormality Detection |
| BMD | 565 | 1,789 | 198 | Cardiac | Heart Sound Classification |
| FluSense | 300 | 376 | 9 | Cough | Flu Detection |
| **Total** | 16,125 | 77,027 | 3,283 | All | Cardiorespiratory Monitoring |

correct responses:

$$\theta^* = \arg\max_\theta \sum_{i=1}^{N} \log p_\theta(\mathbf{y}_i | \mathbf{x}_i, \mathbf{q}_i) \tag{2}$$

This objective is equivalent to minimizing the cross-entropy loss between the model predictions and the target tokens.

We consider instructions spanning seven clinical task categories listed below. These tasks are grounded in established clinical workflows for auscultation (Labkoff et al., 2024; Fallahpour et al., 2025). Examples of these tasks are illustrated in Figure 2.

- Binary Classification: Determining whether sounds are normal or abnormal, whether abnormality (e.g., murmur) is present – the most fundamental screening decision in auscultation

- Detection: Identifying specific acoustic events (e.g., murmurs, crackles) or pathological findings within a recording

- Clinical Reporting: Generating structured summaries of auscultatory findings for documentation in medical records, mirroring standard clinical note-taking

- Diagnostic Reasoning: Explaining the relationship between observed acoustic patterns and underlying pathophysiological mechanisms, with supporting evidence

- Differential Diagnosis: Ranking potential conditions that could explain the observed findings, a key step in clinical decision-making

- Comparative Analysis: Comparing recordings, this task reflects longitudinal monitoring of patients

- Location-Based Analysis: Characterizing sounds at specific anatomical sites (e.g., mitral vs. aortic area), accounting for expected regional variations in normal findings

## 4  StethoBench

We introduce StethoBench, a comprehensive multimodal benchmark for evaluating audio-language models on cardiopulmonary auscultation tasks. StethoBench integrates publicly available datasets covering respiratory sounds, cardiac auscultation, and related conditions, transforming heterogeneous data sources into a unified instruction-response format to enable evaluation across diverse clinical auscultation tasks.

### 4.1  Data Sources.

StethoBench aggregates seven cardiopulmonary audio datasets (Table 1), partitioned into *in-domain* datasets for training, validation, and evaluation, and *out-of-domain* (OOD) datasets for assessing model robustness and generalization.

**In-Domain Datasets:**

- **ICBHI Respiratory Sound Database** (Rocha et al., 2017): 6,898 annotated respiratory cycles from 126 subjects, with labels for crackles, wheezes, and normal breath sounds across diverse auscultation locations.

- **KAUH** (Fraiwan et al., 2021): Respiratory recordings from 112 subjects with comprehensive annotations including diagnosis, adventitious sound types, chest zones, and demographic information.

- **COUGHVID** (Orlandic et al., 2021): Large-scale crowdsourced cough sound dataset with over 25,000 recordings from diverse geographic locations, annotated for COVID-19 status, severity, and respiratory health indicators by expert physicians.

- **CovidUK** (Torresani et al., 2024): UK COVID-19 Vocal Audio Dataset, the largest PCR-referenced collection of respiratory audio recordings with 70,565 linked test results, including cough, exhalation, and speech samples alongside demographic and symptom data.

- **CirCor DigiScope** (Oliveira et al., 2021): The largest pediatric heart sound dataset with 5,282 recordings from 1,568 subjects, featuring detailed murmur characterization (timing, shape, pitch, grading, quality) and manual segmentation of fundamental heart sounds.

- **SPRSound** (Zhang et al., 2022): Pediatric respiratory database containing 2,683 recordings with 9,089 respiratory events from 292 participants, annotated at both record and event levels by experienced pediatric physicians.

- **ZCHSound** (Liu et al., 2024): Pediatric heart sound database from 1,259 participants, including 566 cases of congenital heart disease (CHD) with precise disease diagnoses across multiple CHD categories.

**Out-of-Domain Datasets:**

- **Respiratory@TR** (Altan et al., 2020): RespiratoryDatabase@TR dataset containing multi-channel (12 channels) lung sound recordings from 77 subjects diagnosed with asthma, bronchitis, and five COPD severity grades (COPD0-COPD4), validated by pulmonologists with reference to chest X-rays and pulmonary function tests.

- **CinC** (Liu et al., 2016): PhysioNet/Computing in Cardiology Challenge 2016 dataset, comprising 4,430 heart sound recordings from 1,072 subjects across eight independent databases worldwide, annotated for normal/abnormal classification.

- **BMD-HS** (Ali et al., 2024): BUET Multi-disease Heart Sound dataset with 864 recordings across six diagnostic categories (AS, AR, MR, MS, MD, Normal), featuring multi-label annotations for valvular heart diseases with echocardiographic validation.

- **FluSense** (Al Hossain et al., 2020): Audio dataset from influenza-like illness surveillance platform, containing annotated recordings for nine respiratory event classes (cough, sneeze, sniffle, throat-clearing, speech, breathe, gasp, silence, and other) collected from diverse acoustic environments.

Each dataset includes audio recordings with rich metadata encompassing demographics, clinical diagnoses, auscultation locations, and pathology types. Cardiac datasets provide fine-grained annotations such as murmur characteristics and segmentation boundaries, while respiratory datasets include both record-level diagnoses and event-level annotations for adventitious sounds.

**Out-of-Distribution Definition and Selection.** We define out-of-distribution (OOD) data as recordings that differ from training data in ways that reflect realistic deployment scenarios. These differences encompass multiple types of distribution shift: (1) *Covariate shift*—changes in input distribution $p(\mathbf{x})$ arising from different recording devices and protocols (e.g., TR's 12-channel multi-location recording vs. conventional single-point auscultation), patient populations (e.g., BMD's adult valvular disease patients vs. predominantly pediatric populations in CirCor and ZCHSound training data), or acoustic environments; (2) *Label shift*—changes in outcome distribution $p(y)$ reflecting different disease prevalence across populations; and (3) *Semantic shift*—where the label space or task definition changes, requiring the model to recognize sound categories absent from training (e.g., FluSense's focus on spontaneous respiratory events like sneezes, sniffles, and throat-clearing, which are distinct from the pathological sounds—crackles, wheezes, murmurs—emphasized in clinical training data). This evaluation strategy tests whether domain-specialized training generalizes to the breadth of conditions encountered in clinical practice.

## 4.2 Instruction-Response Pair Generation

To create instruction-response pairs from the audio metadata and labels, we employ a large language model-based synthetic data generation approach (Tang et al., 2023; Hernandez et al., 2024). Specifically, we use LLMs such as GPT-4o (OpenAI, 2024) to generate diverse, clinically grounded instruction-response pairs from structured metadata.

**Single-Audio Analysis.** For each audio recording, we provide the LLM with the ground truth annotations (e.g., diagnosis, location, sound characteristics) and prompt it to generate six distinct instruction-response pairs covering different clinical tasks: (1) binary classification (e.g., normal vs. abnormal), (2) pathology identification, (3) clinical report generation, (4) diagnostic explanation, (5) differential diagnosis, and (6) location-based analysis.

**Pairwise Comparison.** In addition to single-audio analysis, we generate pairwise comparison tasks that require models to identify acoustic differences between two recordings. For each pair of audio recordings with different diagnoses, we prompt the LLM to generate two comparative instruction-response pairs that highlight the key acoustic and diagnostic differences between the recordings. Responses are constrained to a maximum of 40 words to encourage concise, clinically relevant comparisons similar to real clinical discussions.

The complete prompt templates for both single-audio and pairwise tasks are provided in Appendix A. Our approach follows established practices in instruction-tuning literature (Peng et al., 2023; Wang et al., 2022), where GPT-4o generated instruction data has been shown to produce high-quality training signals.

**Quality Control.** To assess the quality and clinical accuracy of generated instruction-response pairs, we manually reviewed a random sample of approximately 2% of the dataset to evaluate factual correctness, clinical relevance, and instruction diversity. For further identification of error types and error estimation, see sythetic data quality analysis in Appendix B.

## 4.3 Dataset Statistics

StethoBench comprises 16,125 audio samples totaling 3,283 minutes (54.7 hours), from which we generated 77,027 instruction-response pairs spanning seven distinct task types (Table 1). The task distribution

reflects clinical workflow diversity: identification tasks (25.9%), classification tasks (21.5%), report generation (16.7%), clinical reasoning (11.8%), differential diagnosis (11.6%), pairwise comparison (4.6%), and location-based analysis (4.6%). Audio recordings vary substantially in duration across datasets, with in-domain recordings averaging between 5.83 seconds (CovidUK) and 22.89 seconds (CirCor), reflecting diverse clinical contexts from brief cough samples to extended cardiac auscultation recordings.

To ensure balanced representation across datasets, we applied sampling strategies to large-scale datasets (e.g., COUGHVID, CovidUK), limiting each dataset's contribution to approximately 1,000-2,000 audio samples while preserving the original train-test splits provided by dataset creators. The final benchmark consists of approximately 13,000 in-domain samples (across 7 datasets) and 2,000 out-of-domain samples (across 4 datasets).

## 5 StethoLM

### 5.1 Model Architecture

StethoLM is a multimodal audio-language model composed of three core modules: an audio encoder $E_A$, a projection network $M_P$, and a language model backbone $G_{\text{LLM}}$.

Given a raw audio waveform $\mathbf{x} \in \mathbb{R}^T$ sampled at 16 kHz, it is first transformed into a mel-spectrogram $\mathbf{s} \in \mathbb{R}^{F \times T'}$ with $F = 64$ frequency bins. The audio encoder $E_A$, an EfficientNet pretrained on medical sounds (Wang et al., 2025), processes the spectrogram to produce a sequence of feature vectors:

$$\mathbf{h}_{\text{audio}} = E_A(\mathbf{s}) \in \mathbb{R}^{N_f \times d_a} \tag{3}$$

where $N_f$ is the number of feature frames and $d_a = 1{,}280$ is the audio feature dimension.

The trainable projection network $M_P$, a multi-layer perceptron, maps these audio features into the language model's embedding space, transforming $\mathbf{h}_{\text{audio}}$ into $k = 4$ prefix tokens, each of dimension $d = 4{,}096$ matching the LLM's hidden dimension:

$$\mathbf{z}_{\text{audio}} = M_P(\mathbf{h}_{\text{audio}}) \in \mathbb{R}^{k \times d} \tag{4}$$

We use MedGemma-4B-IT (Sellergren et al., 2025) as our language model backbone $G_{\text{LLM}}$, chosen for its strong medical domain knowledge. Given an instruction $\mathbf{q}$ and target response $\mathbf{y} = (y_1, \ldots, y_m)$ of length $m$, they are tokenized and embedded to produce $\mathbf{z}_{\text{inst}} \in \mathbb{R}^{n \times d}$ and $\mathbf{z}_{\text{resp}} \in \mathbb{R}^{m \times d}$ respectively, where $n$ is the instruction length. During training, the complete input sequence is:

$$\mathbf{Z} = [\mathbf{z}_{\text{audio}}; \mathbf{z}_{\text{inst}}; \mathbf{z}_{\text{resp}}] \in \mathbb{R}^{(k+n+m) \times d} \tag{5}$$

At inference, the model autoregressively generates response tokens $y_1, \ldots, y_m$ conditioned on $[\mathbf{z}_{\text{audio}}; \mathbf{z}_{\text{inst}}]$ using causal attention.

### 5.2 Training Methodology

We employ a two-stage training strategy: (1) Supervised Fine-Tuning (SFT) to teach the model the primary clinical tasks, followed by (2) an investigation into Direct Preference Optimization (DPO) to refine response quality.

**Supervised Fine-Tuning (SFT).** In the SFT stage, each training sample consists of a triplet $(\mathbf{x}_i, \mathbf{q}_i, \mathbf{y}_i)$ where $\mathbf{y}_i = (y_{i,1}, \ldots, y_{i,m_i})$ is the ground-truth response sequence. The training objective is to maximize the likelihood of response tokens conditioned on the audio and instruction:

$$\mathcal{L}_{\text{SFT}}(\theta) = -\frac{1}{N} \sum_{i=1}^{N} \sum_{j=1}^{m_i} \log p_\theta(y_{i,j} \mid y_{i,<j}, \mathbf{x}_i, \mathbf{q}_i) \tag{6}$$

where $N$ is the number of training samples, $\theta$ denotes all trainable parameters, $y_{i,<j} = (y_{i,1}, \ldots, y_{i,j-1})$ represents preceding response tokens, and the loss is computed only over the $m_i$ response token positions. During

SFT, the parameters of the audio encoder $E_A$ and projection network $M_P$ are fully updated. For the LLM backbone $G_{\text{LLM}}$, we use parameter-efficient fine-tuning with LoRA (Hu et al., 2022), which keeps the original weights frozen and introduces low-rank trainable adapter matrices, significantly reducing computational cost.

**Preference Optimization with mDPO.** Following SFT, we explored direct preference optimization (DPO) (Rafailov et al., 2024) to refine response quality. DPO requires a preference dataset $\mathcal{D}_{\text{pref}} = \{(\mathbf{x}_i, \mathbf{q}_i, \mathbf{y}_{w,i}, \mathbf{y}_{l,i})\}_{i=1}^{N_{\text{pref}}}$, where $\mathbf{y}_w$ is a preferred ("winner") response and $\mathbf{y}_l$ is a dispreferred ("loser") response for the same context $(\mathbf{x}, \mathbf{q})$.

The DPO loss function directly optimizes the policy model $\pi_\theta$ to satisfy the preferences:

$$\mathcal{L}_{\text{DPO}}(\pi_\theta; \pi_{\text{ref}}) = -\mathbb{E}_{(\mathbf{x}, \mathbf{q}, \mathbf{y}_w, \mathbf{y}_l) \sim \mathcal{D}_{\text{pref}}} \left[ \log \sigma \left( \beta \log \frac{\pi_\theta(\mathbf{y}_w \mid \mathbf{x}, \mathbf{q})}{\pi_{\text{ref}}(\mathbf{y}_w \mid \mathbf{x}, \mathbf{q})} - \beta \log \frac{\pi_\theta(\mathbf{y}_l \mid \mathbf{x}, \mathbf{q})}{\pi_{\text{ref}}(\mathbf{y}_l \mid \mathbf{x}, \mathbf{q})} \right) \right] \tag{7}$$

where $\pi_{\text{ref}}$ is a frozen reference model, $\sigma$ is the sigmoid function, and $\beta$ is a temperature parameter controlling the deviation from the reference policy. This objective increases the likelihood of preferred responses $\mathbf{y}_w$ while decreasing that of dispreferred responses $\mathbf{y}_l$.

For multimodal settings, we adopt mDPO (Yu et al., 2024), which extends DPO to model modality-conditional preferences. The key principle is that preference strength should depend on input quality: when audio is clear, the model should confidently distinguish between responses; when audio is degraded, this confidence should decrease appropriately. mDPO achieves this by adding a modality-conditional loss term that encourages the model to assign higher likelihood to the same response when paired with clean audio versus degraded audio.

Concretely, we construct preference pairs by sampling $K = 5$ candidate responses per training sample at varying temperatures ($T \in [0.7, 1.3]$) and ranking them by BERTScore similarity to ground truth (highest as $\mathbf{y}_w$, lowest as $\mathbf{y}_l$). This yields approximately 2400 preference pairs from the training set. During training, we generate degraded audio views $\tilde{\mathbf{x}}$ on-the-fly via temporal cropping, frequency masking, or spectral perturbation. As discussed in Section 7, mDPO did not improve performance despite successful preference learning during training, revealing challenges in applying preference optimization to medical audio-language modeling.

### 5.3 Implementation Details

We train StethoLM on a single NVIDIA H100 GPU using the AdamW optimizer (Loshchilov & Hutter, 2017) with a learning rate of $\eta = 5 \times 10^{-5}$, weight decay $\lambda = 0.01$, and momentum coefficients $\beta = (0.9, 0.98)$. The learning rate is linearly warmed up over the first 100 steps and then held constant. Training is performed for 30 epochs with a batch size of 16, during which the validation loss stabilizes. Model checkpoints are saved based on validation loss, and the best-performing checkpoint is used for evaluation.

Audio inputs are resampled to 16 kHz and segmented into 10-second clips, which are zero-padded or truncated as necessary. For pairwise comparison tasks, we concatenate two 4.5-second audio clips with a 1-second silent gap in-between, maintaining the 10-second total input length. During training, we apply random audio augmentations using Augly (Papakipos & Bitton, 2022). The combined text input (instruction and response) is limited to a maximum of 64 tokens. The audio encoder produces 1,280-dimensional features, which are passed through a 3-layer MLP with a hidden dimension of 2,560 to generate four audio prefix tokens, each of dimension 4,096. We apply LoRA (Hu et al., 2022) to the query (Q), key (K), value (V), and output (O) projection matrices across all attention layers of MedGemma-4B-IT. The LoRA setup uses a rank of $r = 16$, scaling factor $\alpha = 32$, and dropout probability $p = 0.1$. The audio encoder and projection network are trained in full precision (FP32), while the language model is trained with mixed precision (FP16/FP32) for improved memory efficiency.

## 6 Experiments

### 6.1 Evaluation Metrics

Evaluating free-text clinical responses is challenging, as a response can be phrased differently from a reference yet remain clinically correct, or be semantically similar but diagnostically wrong. To address this, we employ two complementary metrics to assess both semantic fidelity and clinical correctness.

**BERTScore F1.** We use BERTScore (Zhang et al., 2019) to measure semantic similarity between generated responses and ground-truth references. BERTScore computes token-level similarities using contextual embeddings from a pretrained language model, providing a more robust measure of semantic equivalence than n-gram-based metrics like BLEU or ROUGE. We use the F1 formulation, which balances precision (how much of the generation matches the reference) and recall (how much of the reference is covered by the generation). We employ DeBERTa-large-mnli (He et al., 2021) as the embedding model. DeBERTa's enhanced attention mechanism and larger model capacity provide superior semantic understanding, particularly for domain-specific technical language. The MNLI (Multi-Genre Natural Language Inference) fine-tuning further improves the model's ability to assess semantic relationships in diverse contexts, making it well-suited for evaluating medical text where precise semantic distinctions matter—for instance, distinguishing between "systolic murmur" and "diastolic murmur" or "early inspiratory crackles" versus "late inspiratory crackles." We report BERTScore as percentages (0-100%).

**Clinical Accuracy. (LLM-Judged).** While BERTScore excels at measuring semantic similarity, it can penalize clinically valid responses that use synonyms (e.g., "basilar crepitations" vs. "crackles at the lung bases") or fail to detect responses that are fluent but factually incorrect. To directly evaluate diagnostic validity/accuracy, we measure *Clinical Accuracy* using a state-of-the-art large language model, GPT-5 (OpenAI, 2025), as an automated evaluator. For each generated response, the LLM judge is provided with both the model's output and the ground-truth reference. It is then prompted to make a binary determination (*yes/no*): is the generated response clinically consistent with the reference? The prompt (see Appendix C) is designed to ignore minor stylistic variations and focus strictly on whether the core clinical information is correctly conveyed. We report accuracy as the percentage of responses judged to be correct (i.e., *yes*). This metric serves as a crucial proxy for diagnostic utility in real-world clinical applications. To validate robustness and add transparency to the current metric, we conducted sensitivity analyses across multiple judge models and prompt variations (Appendix D).

**Metric Selection and Complementarity.** We prioritize BERTScore in main results (with ROUGE-1 and METEOR in Appendix F) because its contextualized embeddings better capture semantic equivalence in medical language than n-gram metrics—recognizing, for instance, that "basilar crepitations" and "crackles at the lung bases" convey the same clinical meaning. This approach aligns with recent medical multimodal models (Sellergren et al., 2025; Fallahpour et al., 2025) and provides continuous measurement of partial correctness. However, semantic similarity does not guarantee diagnostic validity. A response can be fluent and semantically coherent while containing clinical errors—for example, correctly identifying abnormality (high BERTScore) but misclassifying the specific pathology (low accuracy). We therefore report both BERTScore and clinical accuracy as complementary metrics: BERTScore captures linguistic and semantic alignment, while LLM-judged accuracy evaluates diagnostic correctness.

### 6.2 Baseline Models

We compare StethoLM against both general-purpose audio-language models and a text-only configuration to assess model performance and training effectiveness. All baselines are evaluated using identical experimental protocols and metrics.

**Text-Only LLM Baseline.** We evaluate MedGemma-4B-IT (Sellergren et al., 2025), the same language model backbone used in StethoLM, receiving only text instructions without audio input. This configuration is critical for interpreting BERTScore results, as contextual embeddings can yield relatively high scores

Table 2: **Performance comparison on the StethoBench benchmark.** BertS denotes BERTScore and Acc denotes Accuracy (both in %). The best score for each metric per task is highlighted in **bold**, and the second-best score is underlined. *Note:* "Gemini" refers to **Gemini-2.5-Flash** and "Qwen" refers to **Qwen2.5-Omni**. Additional baseline comparisons with Qwen3-Omni (Xu et al., 2025b) and Audio Flamingo 3 (Goel et al., 2025) are provided in Appendix E. Supplementary metrics (ROUGE-1, METEOR) and relaxed accuracy evaluation for DDx focusing on clinical plausibility are provided in Appendix F.

| Task | Models | | | | | | | | | | | | | | |
| --- | --- | --- | --- | --- | --- | --- | --- | --- | --- | --- | --- | --- | --- | --- | --- |
| | LLM | | Pengi | | LTU | | Gama | | Gemma3N | | Gemini | | Qwen | | StethoLM | |
| | BertS | Acc | BertS | Acc | BertS | Acc | BertS | Acc | BertS | Acc | BertS | Acc | BertS | Acc | BertS | Acc |
| Classification | 48.6 | 4.9 | 33.6 | 7.6 | 49.0 | 2.5 | 46.4 | 7.5 | 51.3 | 31.3 | 49.3 | 49.5 | 58.9 | 31.0 | **75.5** | **66.4** |
| Detection | 43.0 | 2.8 | 30.8 | 4.3 | 43.1 | 3.6 | 44.0 | 4.2 | 47.1 | 11.5 | 45.9 | 19.2 | 52.5 | 22.0 | **70.4** | **47.9** |
| Reporting | 41.1 | 2.2 | 30.7 | 4.1 | 45.1 | 2.0 | 44.7 | 2.0 | 47.5 | 9.2 | 49.8 | 13.2 | 56.7 | 12.7 | **72.8** | **36.2** |
| Reasoning | 41.8 | 0.0 | 27.6 | 1.0 | 43.7 | 4.0 | 46.5 | 5.3 | 45.2 | 9.0 | 47.6 | 22.2 | 55.6 | 29.0 | **71.4** | **44.1** |
| DDx | 42.3 | 3.3 | 26.6 | 4.1 | 44.9 | 4.0 | 44.6 | 4.0 | 44.5 | 3.7 | 43.6 | 13.2 | 60.2 | 19.8 | **67.7** | **30.6** |
| Comparison | 45.1 | 2.3 | 28.4 | 1.7 | 47.2 | 2.1 | 45.2 | 1.9 | 47.7 | 12.7 | 46.9 | 15.4 | 57.8 | 22.7 | **70.7** | **40.5** |
| Location | 44.4 | 4.3 | 26.4 | 1.1 | 46.4 | 4.4 | 46.0 | 2.2 | 49.0 | 11.0 | 45.6 | 14.3 | 54.6 | 11.3 | **72.0** | **36.8** |
| **Overall** | 43.8 | 2.8 | 29.2 | 3.4 | 45.6 | 3.2 | 45.3 | 3.9 | 47.5 | 12.6 | 47.0 | 21.0 | 56.5 | 21.2 | **71.8** | **47.8** |

($\approx$ 40+) even without audio information when the model generates fluent, medically plausible text based solely on instruction context.

**General Audio-Language Models.** We compare against three audio-language models trained on general audio: Pengi (Deshmukh et al., 2023), an audio-grounded language model using audio embeddings as soft prompts for a frozen language model; LTU (Gong et al., 2023), which integrates pretrained audio encoders with instruction-tuned LLMs; GAMA (Ghosh et al., 2024), a generative audio-language model using audio prefix tokens; and Gemma3N (Google DeepMind, 2025), a lightweight multimodal (audio–image–video–text) instruction-tuned model that processes audio via native audio tokens rather than external encoders. These models represent state-of-the-art general audio understanding but lack specialized training on cardiopulmonary sounds.

**Large Multimodal Models.** We evaluate two foundation models with multimodal capabilities: Qwen2.5-Omni (Xu et al., 2025a), an open-source audio-language model supporting diverse audio understanding tasks, and Gemini-2.5-Flash (Comanici et al., 2025), Google's multimodal model with audio, video, image, and text understanding. These models with tens to hundreds of billions of parameters, trained on web-scale multimodal data, represent a fundamentally different approach than specialized audio-language models. Rather than being optimized specifically for audio-text tasks, they are generalist systems designed to handle any combination of modalities. Comparing against these models reveals whether domain-specialized training on medical audio can compete with or surpass the emergent capabilities of large-scale general-purpose systems.

### 6.3 In-Domain Performance

We evaluate all models on StethoBench's test set containing approximately 20,000 instruction-response pairs spanning seven clinical task categories (Table 2). Performance is measured using both BERTScore and ground truth-generation match accuracy, providing complementary views of generation quality and clinical correctness. Additional evaluation using ROUGE-1 and METEOR metrics is provided in Table 9, Appendix F.

**StethoLM Achieves Substantial Gains Across All Tasks.** StethoLM attains an overall BERTScore of 71.8% and accuracy of 47.8%, outperforming all baseline models (Table 2). Relative to the strongest baseline, Qwen2.5-Omni (56.5% BERTScore, 21.2% accuracy), StethoLM yields absolute improvements of +15.3 percentage points in BERTScore and +26.6 percentage points in accuracy. The model demonstrates

superior performance across all seven task categories, with notably strong results on classification (75.5% BERTScore, 66.4% accuracy) and detection tasks (70.4%, 47.9%). These consistent performance gains across task types ranging from binary classification to complex clinical reasoning provide evidence that domain-specialized training on cardiopulmonary audio yields robust benefits beyond those achievable through general-purpose audio understanding or large-scale pretraining alone.

**Audio Grounding Is Essential for Clinically Meaningful Understanding.** The text-only baseline (LLM) attains a moderate BERTScore (43.8%) despite minimal accuracy (2.8%), indicating that linguistic fluency and medical knowledge alone are insufficient for clinically valid reasoning. MedGemma-4B-IT, pretrained extensively on medical text, can produce coherent responses containing appropriate terminology even without access to the audio signal. However, without acoustic grounding, these responses remain generic and fail to reflect the true respiratory and cardiac findings. This contrast highlights that high textual similarity does not equal diagnostic correctness, and incorporating the clinical accuracy metric is necessary.

**Task Performance Reflects Inherent Complexity and Evaluation Challenges.** Performance varies substantially across task categories, revealing systematic differences in task difficulty (see Table 2). Binary classification tasks achieve the highest accuracy (66.4%) with relatively small BERTScore-accuracy discrepancies (9.1 points), consistent with their objective nature and unambiguous ground truth. Detection (70.4% BERTScore, 47.9% accuracy), reporting (72.8%, 36.2%), and reasoning tasks (71.4%, 44.1%) exhibit moderate accuracy with larger metric divergence. These generative tasks require selecting and synthesizing information: for reporting, the model must choose which acoustic findings to emphasize; for reasoning, which pathophysiological mechanisms to elaborate. High BERTScore indicates the model generates clinically appropriate content, while lower accuracy reflects variation in how information is organized and emphasized compared to reference annotations synthesized from human labels. Differential diagnosis presents the greatest challenge (67.7% BERTScore, 30.6% accuracy). Beyond identifying relevant conditions, this task requires ranking them by likelihood, which is a process with inherent subjectivity. We include this challenging task because differential diagnosis represents essential clinical reasoning. This performance pattern mirrors documented variation in human diagnostic agreement. Studies in diagnostic pathology show that inter-rater concordance varies systematically with task ambiguity: pathologists achieve 87–96% agreement on objective classifications (benign vs. invasive carcinoma) but only 48–84% on ambiguous intermediate diagnoses (Elmore et al., 2016). Our results follow this pattern, with accuracy declining as tasks shift from objective classification to subjective prioritization. The BERTScore-accuracy divergence thus reflects both model limitations and inherent task subjectivity: the model generates semantically appropriate clinical content (captured by BERTScore) while showing incomplete alignment with specific reference formulations (measured by accuracy).

**Domain Specialization Proves Essential for Clinical Audio Understanding.** Models trained on general-purpose audio struggle significantly, with general audio-language models (Pengi, LTU, GAMA) achieving BERTScores of 29–45% and accuracy below 4%. This poor transfer indicates that representations learned from music, environmental sounds, and speech emphasize acoustic features fundamentally different from those critical for auscultation. In medical audio, diagnostically relevant information resides in subtle temporal patterns (crackle duration: <5ms vs >10ms distinguishes fine vs coarse), spectral characteristics (wheeze pitch: monophonic vs polyphonic), and phase-specific variations (systolic vs diastolic timing). These features are not present in or relevant to general audio tasks.

Large-scale multimodal models provide some benefit through massive scale and diverse pretraining. Gemini-2.5-Flash (47.0% BERTScore, 21.0% accuracy) and Qwen2.5-Omni (56.5% BERTScore, 21.2% accuracy) substantially outperform general audio-language models despite lacking medical audio training. Qwen2.5-Omni, with explicit audio-language training on diverse audio data, approaches competitive BERTScore performance. However, both models fall substantially short of StethoLM in accuracy (∼21% vs 47.8%), demonstrating that domain-specialized training on cardiopulmonary audio provides irreplaceable benefits that cannot be fully compensated by scale alone. This finding aligns with recent evidence that medical foundation models require domain-specific data to achieve clinical-grade performance (Singhal et al., 2023; Moor et al., 2023).

Table 3: **Out-of-Domain (OOD) performance comparison across datasets.** BertS denotes BERTScore and Acc denotes Accuracy (both in %). The best score for each metric per dataset is highlighted in **bold**, and the second-best score is underlined. Additional baseline comparisons with Qwen3-Omni (Xu et al., 2025b) and Audio Flamingo 3 (Goel et al., 2025) are provided in Appendix E. Supplementary metrics (ROUGE-1, METEOR) are provided in Appendix F.

| Dataset | Models | | | | | | | | | | | | | | | |
|---|---|---|---|---|---|---|---|---|---|---|---|---|---|---|---|---|
| | LLM | | Pengi | | LTU | | Gama | | Gemma3N | | Gemini | | Qwen | | StethoLM | |
| | BertS | Acc | BertS | Acc | BertS | Acc | BertS | Acc | BertS | Acc | BertS | Acc | BertS | Acc | BertS | Acc |
| TR | 44.0 | 0.5 | 27.7 | 1.6 | 44.2 | 0.6 | 44.0 | 0.3 | 48.4 | 5.6 | 44.8 | 7.5 | 60.7 | 17.7 | **66.2** | **25.7** |
| CinC | 39.5 | 1.1 | 29.4 | 1.3 | 42.3 | 0.4 | 42.6 | 0.7 | 44.5 | 5.2 | 45.9 | 21.5 | 54.0 | 12.4 | **63.3** | **22.2** |
| BMD | 45.0 | 2.5 | 27.3 | 1.4 | 47.2 | 1.2 | 46.7 | 4.4 | 48.2 | 11.7 | 40.7 | 20.9 | 58.6 | 17.3 | **67.3** | **30.4** |
| FluSense | 45.8 | 0.3 | 30.6 | 15.6 | 45.8 | 9.4 | 47.3 | 22.3 | 50.7 | 9.4 | 52.3 | 14.3 | 59.4 | **37.1** | **61.5** | 23.2 |
| **Overall** | 43.6 | 1.1 | 28.8 | 5.0 | 44.9 | 2.9 | 45.2 | 6.9 | 48.0 | 8.0 | 45.9 | 16.1 | 58.2 | 21.1 | **64.8** | **25.2** |

## 6.4 Out-of-Domain Generalization

We evaluate StethoLM's robustness on four external datasets: RespiratoryDatabase@TR (TR), CinC2016, BMD-HS (BMD), and FluSense, testing generalization across different recording devices, acoustic environments, and patient populations (see Table 3). StethoLM achieves 64.8% BERTScore and 25.2% accuracy on out-of-domain data, representing a moderate BERTScore decline (7.0 percentage points) but substantial accuracy decrease (22.6 percentage points) relative to in-domain performance (71.8%, 47.8%). This divergence reveals partial generalization: the model maintains semantic understanding of clinical language and acoustic features but struggles with precise diagnostic mappings under distribution shift. Additional OOD evaluation using ROUGE-1 and METEOR is provided in Table 10, Appendix F.

StethoLM achieves the best overall performance across OOD datasets and maintains superiority on three of four benchmarks. The model performs optimally on TR (66.2% BERTScore, 25.7% accuracy), CinC (63.3%, 22.2%), and BMD-HS (67.3%, 30.4%), which feature high-quality recordings from controlled clinical settings with rigorous quality control protocols (Altan et al., 2020; Ali et al., 2024). However, Qwen2.5-Omni surpasses StethoLM on FluSense accuracy (37.1% vs 23.2%), revealing critical limitations of domain-specialized training when confronted with challenging real-world conditions. FluSense differs fundamentally from clinical datasets in capturing spontaneous respiratory events (sneezes, sniffles, throat-clearing, gasps) that are absent from structured clinical auscultation, where protocols focus on controlled breathing cycles (Al Hossain et al., 2020). This creates a distribution mismatch: StethoLM's training provides little supervision for these everyday respiratory behaviors, while Qwen2.5-Omni's diverse general audio training encompasses such spontaneous events. Additionally, FluSense recordings exhibit environmental noise, variable quality from crowdsourced collection. Detailed failure analysis examining out-of-vocabulary sound events is provided in Appendix G.

## 6.5 Zero-Shot Classification

Beyond instruction-following tasks, we investigate whether StethoLM's learned representations can support zero-shot classification, i.e. predicting class labels the model has never been trained on. To do this, we first generate a descriptive report for each audio input using StethoLM. We then encode this report using an off-the-shelf sentence encoder (text-embedding-v4 from Qwen3-Embedding (Zhang et al., 2025)) and compute cosine similarities with embeddings of all candidate class labels. The audio is assigned the label with highest similarity to its generated report. Accuracy is computed by comparing predicted labels against ground truth.

We compare StethoLM against AudioMAE (Huang et al., 2022), CLAP (Wu et al., 2023), and Opera-CE (Zhang et al., 2024a), three representative pre-trained audio models that produce fixed-dimensional embeddings optimized for downstream classification. In contrast, StethoLM is trained solely for generative

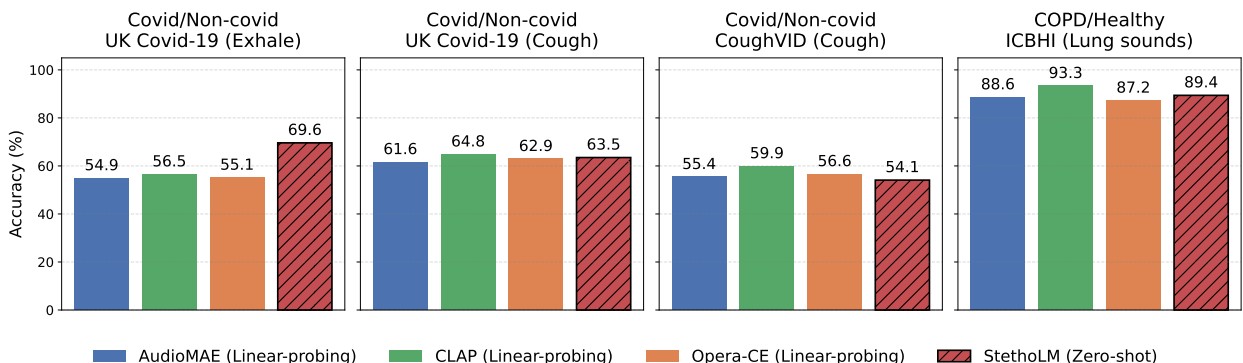

Figure 3: **Zero-shot classification comparison with classification models.** Accuracy across four binary tasks comparing zero-shot inference (StethoLM via report generation + text similarity) against supervised linear probing (AudioMAE, CLAP, Opera-CE). Tasks include COVID-19 detection from exhalation/cough recordings and COPD screening from lung sounds.

instruction-following without explicit classification objectives. Figure 3 presents zero-shot accuracy across four classification tasks.

StethoLM demonstrates competitive zero-shot classification performance. Notably, StethoLM excels on exhalation-based COVID detection (69.6% vs. 56.5% CLAP), demonstrates competitive performance on cough-based COVID detection (63.5% vs. 64.8% CLAP), and achieves strong results on COPD detection (89.4% vs. 93.3% CLAP). These results suggest that generative instruction-tuning captures clinically meaningful acoustic features transferable to classification tasks, despite no explicit classification training.

## 6.6 Audio Grounding and Model Explainability

We analyzed attention patterns when StethoLM generates responses to clinical instructions paired with cardiopulmonary audio on the test set. The heatmap (see Figure 4) visualizes the attention matrix for layer 3, head 2, showing how query tokens attend to key tokens across three distinct regions: audio embeddings (4 tokens), instruction text (14 tokens), and generated response (20 tokens). Black solid lines separate the audio region from instruction tokens, while red dashed lines mark where response generation begins. The pattern reveals that generated response tokens maintain substantial attention to the audio prefix throughout generation, confirming that the model actively processes acoustic information during clinical reasoning at inference time. The attention weights show clear differentiation between the three input modalities, with significant attention from response tokens back to the audio embeddings.

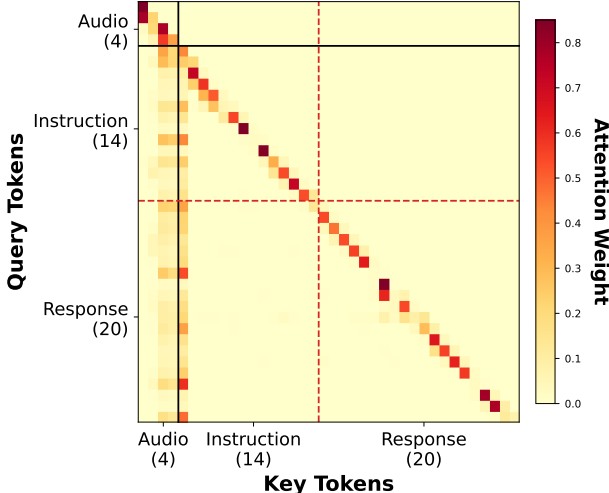

Figure 4: **Attention patterns revealing audio grounding during response generation.** Heatmap visualizes how generated response tokens (vertical axis) attend to key tokens (horizontal axis). Black solid lines mark the audio-instruction boundary; red dashed lines mark where response generation begins.

| Inference Query | ✓ Ground Truth | StethoLM Generation |
|---|---|---|
| Classify this lung sound as normal or abnormal. | Abnormal. | Abnormal. The presence of adventitious sounds suggests a respiratory pathology. |
| What type of sound is this? | The patient likely has covid-19 based on the exhalation sound. | The patient likely has covid-19 based on the exhalation pattern. |
| Generate a brief summary of the lung sound analysis. | The recording presents with possible wheezing. This suggests potential airway constriction. | The recording presents with possible wheezing, suggesting airway constriction. |
| Explain the reasoning behind the diagnosis of asthma. | Asthma is characterized by reversible airway obstruction and bronchial hyper-responsiveness, which can manifest as wheezing. | Asthma is indicated by wheezing, a common finding in obstructive airway diseases. |
| List three differential diagnoses for this cough | 1. Upper respiratory infection. 2. Acute bronchitis. 3. Irritant cough (e.g., from smoke or allergens). | 1. Upper respiratory infection. 2. Acute bronchitis. 3. Irritant cough. |
| Which audio would prompt you to order an echocardiogram, and why? | Audio 2, suggestive of VSD, warrants an echocardiogram to confirm the diagnosis and assess the defect's size and hemodynamic impact. | Audio 2, as it suggests a ventricular septal defect, potentially requiring further cardiac evaluation with an echocardiogram. |
| Based on the posterior right location, what findings would support a diagnosis of COPD? | Expected findings include diminished breath sounds, possible wheezes (especially on expiration). | The presence of wheezes or prolonged expiration in the posterior right lung field would support a diagnosis of COPD. |

Figure 5: **Qualitative analysis of StethoLM's clinical reasoning capabilities.** Representative outputs demonstrate the model's ability to generate clinically coherent responses across diverse auscultation tasks.

## 6.7 Qualitative Analysis

To complement quantitative metrics, we examine representative examples of StethoLM's responses across seven clinical task categories (see Figure 5). The model consistently generates clinically coherent responses that preserve diagnostic validity while often extending beyond ground-truth references with medically appropriate context. For instance, when classifying sounds as abnormal, the model correctly identifies abnormality and supplements with clinical reasoning ("The presence of adventitious sounds suggests a respiratory pathology"). Explanation tasks reveal integration of acoustic observations with pathophysiological knowledge ("characterized by reversible airway obstruction and bronchial hyper-responsiveness"). While these examples demonstrate strong performance, we also conducted systematic failure case analysis to identify key limitations and improvement directions (Appendix G). Overall, StethoLM's behavior suggests suitability for clinical decision support rather than autonomous diagnosis, with its medical coherence and contextual reasoning valuable for augmenting clinician judgment while requiring expert supervision for high-stakes diagnostic scenarios.

# 7 Ablations

To understand which design choices contribute most to StethoLM's performance, we conducted systematic ablation studies by modifying individual components while keeping all others fixed. We evaluate each variant on both semantic similarity (BERTScore F1) and clinical accuracy (GPT-5 judged) to understand their complementary effects.

## 7.1 Component Ablations

**Audio Contribution.** Removing audio input at inference results in substantial performance degradation (BERTScore: 71.8% → 57.8%; accuracy: 47.8% → 28.5%), confirming that acoustic information is essential for clinical audio understanding tasks. To further validate this finding, we conduct an additional sanity check

by shuffling the audio modality to create text-audio mismatch (Madaan et al., 2025), where each instruction receives a randomly selected audio recording from a different sample in the test set. This maintains the input modality structure while breaking the semantic correspondence between audio and instruction. Under shuffled pairing, performance remains substantially degraded (BERTScore: 71.8% → 63.6%; accuracy: 47.8% → 31.9%), confirming that the model relies on meaningful audio-text correspondence rather than exploiting text-only patterns. Notably, while the base medical language model (MedGemma-4B-IT) appropriately refuses to answer audio-related questions without audio input (2.8% accuracy in Table 2), the multimodally-adapted model continues generating clinical responses even when audio is absent or mismatched. This behavior reflects a broader phenomenon in multimodal adaptation where safety mechanisms can be inadvertently compromised (Lee et al., 2024). While detailed safety analysis is beyond this work's scope, this observation highlights the importance of preserving refusal behaviors in clinical AI systems.

**Audio Encoder Selection.** We evaluate alternative audio encoders while maintaining all other architectural components. Both CLAP (71.4% BERTScore, 47.0% accuracy) and AST (71.8%, 47.5%) achieve performance comparable to the default encoder (71.8%, 47.8%). This robustness across encoder architectures suggests that the benefits of domain-specialized training are largely independent of the specific audio encoding method, provided the encoder captures sufficient temporal and spectral information.

**Language Model Backbone.** The choice of language model backbone substantially impacts performance. The default medical-specialized backbone achieves 71.8% BERTScore and 47.8% accuracy. Llama-3.2-3B-Instruct, a general-purpose model, attains 70.4% BERTScore and 45.3% accuracy, representing modest degradation. MediPhi-Instruct, despite medical pretraining, yields 68.5% BERTScore and 43.4% accuracy, while the smaller DeepSeek-R1-Distill-Qwen-1.5B achieves 66.7% and 39.4%. These results indicate that both model capacity and medical domain knowledge contribute to performance, with larger models demonstrating greater capacity to integrate audio-language representations even without extensive medical pretraining.

**Prefix Length Impact.** We vary the number of audio prefix tokens (k) to assess the trade-off between expressiveness and efficiency. The default configuration (k=4) achieves 71.8% BERTScore and 47.8% accuracy. Reducing to k=2 yields 70.7% and 45.0%. Increasing to k=8 attains 71.5% and 47.5%, showing minimal difference. These results indicate that k=4 provides an effective balance, with longer prefixes offering diminishing returns while shorter prefixes compromise representational capacity.

## 7.2 Challenges in Preference Optimization

We investigated multimodal direct preference optimization (mDPO) using on-policy sampling, where multiple candidate responses were generated at different temperatures and ranked by BERTScore similarity to ground truth annotations. Training metrics indicated successful preference learning: the model achieved 78% accuracy in preferring chosen over rejected responses and 71% accuracy in assigning higher likelihood to responses paired with clean versus degraded audio. However, evaluation on the test set showed slight performance degradation (BERTScore: 71.8% → 69.8%; accuracy: 47.8% → 45.9%).

This failure likely reflects limitations in using BERTScore as a preference signal and challenges inherent to preference optimization in this domain. BERTScore captures semantic similarity and linguistic fluency but does not reliably reflect diagnostic correctness. It may favor responses that are generically plausible yet clinically imprecise (missing severity qualifiers, overgeneralizing findings, omitting anatomical/temporal specificity). Additionally, supervised fine-tuning on 70k instruction-response pairs already provides strong supervision for audio-text alignment and clinical reasoning, potentially leaving limited headroom for preference-based refinement.

## 8 Discussion

StethoLM demonstrates that instruction-following audio-language models can successfully perform diverse clinical reasoning tasks in cardiopulmonary auscultation. Our results reveal three critical insights that advance understanding of multimodal medical AI. First, audio grounding is essential for clinical reasoning. The

Table 4: **Ablation study on StethoLM components.** Each variant modifies one component of the StethoLM configuration. BertS denotes BERTScore (%), and Acc denotes accuracy (%). All models are evaluated on the StethoBench benchmark.

| Component | Configuration | BertS. | Acc. |
|---|---|---|---|
| **StethoLM** | *Full model: Audio+Text, all trainable* | **71.8** | **47.8** |
| Modality | Audio removed at inference | 57.8 | 28.5 |
| | Audio-text mismatch (shuffled pairing) | 63.6 | 31.9 |
| Audio Encoder | CLAP | 71.4 | 47.0 |
| | AST | 71.8 | 47.5 |
| LLM Backbone | Llama-3.2-3B-Instruct | 70.4 | 45.3 |
| | MediPhi-Instruct | 68.5 | 43.4 |
| | DeepSeek-R1-Distill-Qwen-1.5B | 66.7 | 39.4 |
| Prefix Length | k=2 tokens | 70.7 | 45.0 |
| | k=8 tokens | 71.5 | 47.5 |
| Preference Optimization | + Multimodal DPO (mDPO) | 69.8 | 45.9 |

improvement in BERTScore (71.8% vs 43.8%) and in accuracy (47.8% vs 2.8%) when audio is incorporated demonstrates that acoustic information provides irreplaceable diagnostic evidence. Second, domain specialization proves essential. StethoLM's improvement over the strongest baseline (Qwen2.5-Omni) demonstrates that representations learned from general audio, e.g., music, speech, environmental sounds, do not capture diagnostically relevant features in cardiopulmonary auscultation, such as temporal characteristics distinguishing fine versus coarse crackles. Third, task characteristics fundamentally shape evaluation. The model excels at objective classification (66.4% accuracy) but struggles with differential diagnosis (30.6%), reflecting not just model limitations but the inherent subjectivity of certain clinical tasks where expert disagreement is common (Elmore et al., 2016).

**Clinical Decision Support: Capabilities and Constraints.** StethoLM's ability to generate natural language explanations alongside diagnostic predictions addresses a critical limitation of existing auscultation AI systems that produce only fixed categorical outputs. This interpretability enables several clinical applications that extend beyond narrow classification. For triage and screening, the model can process recordings to flag potential abnormalities with accompanying descriptions, supporting rapid initial assessment in high-volume settings. For longitudinal monitoring, the model's comparison capabilities allow tracking disease progression through serial recordings. However, our results reveal significant constraints that preclude autonomous diagnostic deployment. An overall accuracy of 47.8% and the model's performance degradation on out-of-domain data (25.2% accuracy) presents challenges for real-world deployment. Most concerning is the safety implication we observed during ablation studies: after multimodal training, the model generates diagnostic responses even when audio input is absent. This poses risks in clinical settings where sensor failures or data corruption could result in plausible but incorrect output (Li et al., 2023). These findings suggest a viable clinical role for StethoLM as augmentative decision support rather than autonomous system. In this paradigm, the model generates candidate interpretations and supporting reasoning that expert clinicians review, verify, and refine. However, this augmentative role still requires prospective clinical validation to assess real-world utility, measure impact on clinician workflow, and identify potential failure modes in deployment settings.

**Limitations.** We acknowledge several limitations that constrain interpretation and generalization of our findings. First, our evaluation relies on single reference annotations, which may penalize valid alternatives in subjective tasks such as differential diagnosis ranking or clinical description generation. Second, StethoBench aggregates datasets of heterogeneous quality, annotation standards, and demographic coverage, with some geographic and population biases and variable recording conditions, which may affect performance and gener-

alizability. Thirdly, our evaluation relies on LLM-based judges for clinical accuracy assessment, which, while robust across multiple models and prompts, lacks expert clinician validation. Expert adjudication would be particularly valuable for ambiguous diagnostic tasks and would help calibrate performance expectations against human baselines. Finally, multimodal training can compromise refusal behavior, with the model occasionally producing responses without audio input rather than declining appropriately. These limitations indicate that while StethoLM advances instruction-following AI for clinical auscultation, further work on real-world evaluation and safety validation is necessary before clinical application.

## 9    Conclusion

We presented StethoLM, the first audio-language model specialized for cardiopulmonary auscultation that performs instruction-driven clinical reasoning across the full spectrum of auscultation tasks. Through multi-stage training on StethoBench, our comprehensive benchmark of 77,027 synthesized instruction-response pairs, StethoLM achieves substantial improvements over general-purpose audio-language models, demonstrating that domain-specialized training on clinical audio enables robust acoustic reasoning. Our key contributions include: (1) StethoLM's architecture and training methodology for audio-language understanding in clinical auscultation; (2) StethoBench, a diverse benchmark spanning seven clinical task categories; (3) comprehensive evaluation revealing both capabilities and limitations (e.g., challenges in differential diagnosis ranking, sensitivity to recording quality).

**Future Directions.**    We identify several promising directions emerge from this work. First, multimodal integration: extending StethoLM to incorporate patient demographics and medical history could enable more holistic diagnostic reasoning that mirrors clinical practice. Second, interactive refinement: allowing clinicians to iteratively query the model ("Describe the timing of this sound," "Could this be pneumonia instead?") would support exploratory clinical reasoning. Third, uncertainty quantification: explicit modeling of epistemic uncertainty would better calibrate the model's confidence and flag cases requiring expert review. Finally, prospective clinical validation: real-world deployment studies are essential to assess clinical utility, safety, and integration into existing workflows.

StethoLM establishes a foundation for instruction-following AI systems in clinical auscultation, demonstrating that audio-language models can move beyond narrow classification to support the diverse reasoning tasks clinicians perform. By combining domain-specialized training with interpretable natural language interfaces, this work takes a step toward more general, trustworthy, and clinically useful medical AI. As cardiopulmonary diseases remain among the leading causes of mortality worldwide, scalable AI tools that possess clinicians' diagnostic capabilities hold significant promise for improving global health outcomes.

## Acknowledgments

This work was supported by the NWO AiNed Fellowship Grant awarded to A.S., and in part by Google.org and the Google Cloud Research Credits program through the Gemini Academic Program. We also acknowledge the use of the Dutch National Supercomputer Snellius for essential computational tasks.

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

# A   Instruction-Response Generation Prompts

Below are the complete system prompts used to generate instruction-response pairs from cardiopulmonary audio metadata (See Section 4.2 for details about data generation).

## A.1   Single-Audio Analysis

---

**LLM prompt for single-audio pair generation.**

You are a medical expert specializing in cardiopulmonary auscultation.

Based on the following ground truth for a lung sound recording: ['diagnosis':row['diagnosis'], 'location': row['location']], generate 6 diverse instruction-response pairs that simulate realistic clinical tasks.

Each instruction should reflect a distinct clinical task such as:

```
1. Classifying audio as normal or abnormal
2. Identifying the likely pathology
3. Generating a summary or report
4. Providing diagnosis and explanation
5. Differential diagnosis (What could be causing this?)
6. Location-based analysis
```

Return the output as a JSON list with the following structure:

```
[
  {
    "instruction": "What pathology is suggested by this audio?",
    "response": "Pneumonia."
  },
  {
    "instruction": "What is the primary finding in this lung sound?",
    "response": "Crackles are present."
  },
  ...
]
```

Make sure each pair is distinct in focus and clinically grounded in the diagnosis row['diagnosis']. Be concise and clinically accurate.

---

## A.2   Pairwise Comparison Prompt

---

**LLM prompt for pairwise comparison data generation.**

You are a medical expert specializing in lung sound interpretation.

You are comparing two anonymized lung sound recordings. Each has associated ground truth information:

Audio 1:

```
Diagnosis: {row['diagnosis_1']}
```

Audio 2:

```
Diagnosis: {row['diagnosis_2']}
```

Generate two realistic instruction-response pairs simulating real clinical comparative analysis. Instructions should be phrased as questions a doctor or medical trainee may ask. The response should be concise (max 40 words).

Example JSON output and format:

```
[
 {
   "instruction": "How do the sounds in Audio 2 differ from those in Audio 1?",
   "response": "Audio 2 features coarse crackles suggestive of pneumonia, while
   Audio 1 has no abnormal sound."
 },
 {
   "instruction": "Compare Audio 2 to Audio 1. What is the key acoustic difference?",
   "response": "Audio 2 features wheezes."
 }
]
```

Make sure comparisons are clinically relevant and accurately reflect the diagnostic differences between the two recordings.

---

# B    Synthetic Data Quality Analysis

To validate the clinical reliability of our LLM-generated instruction-response pairs (Section 4.2), we conducted manual review to characterize error patterns and assess benchmark quality. This appendix details the methodology, observed patterns, and their implications.

## B.1    Validation Methodology

One author with clinical training conducted a two-phase review. First, an exploratory phase examined multiple samples across task types to identify systematic error patterns. Second, a structured validation phase reviewed 10 randomly sampled instruction-response pairs per task type (70 total pairs) to characterize error frequency and severity. For each pair, the reviewer assessed: (1) *Factual consistency:* Does the response accurately reflect the ground-truth annotations (diagnosis, location, sound characteristics)? (2) *Clinical plausibility:* Is the response medically reasonable given the available information? (3) *Task appropriateness:* Does the response match the instruction's requirements?

Samples were categorized as:

- **Accurate**: fully consistent with ground truth,

- **Acoustic Hallucination**: adds sound features not in annotations,

- **Clinical Over-Elaboration**: adds plausible but unsupported clinical interpretations, or

- **Contradictions**: factually inconsistent with ground truth.

## B.2    Representative Examples

Table 5: **Examples of error patterns in synthetic instruction-response generation.**

| Ground Truth | Instruction | Generated Response | Error Type |
|---|---|---|---|
| Diagnosis: Bronchitis, Sound: Wheezing | What could be the differential diagnosis? | While the primary diagnosis is bronchitis, differentials could include asthma or pneumonia, given obstructive sounds like wheezing and *rhonchi.* | Acoustic hallucination (rhonchi added) |
| Diagnosis: Pneumonia (severe), Sounds: Crackles | Explain your reasoning for suspecting severe pneumonia. | The presence of coarse crackles, *consistent with fluid accumulation and alveolar collapse*, strongly suggests severe pneumonia. | Clinical over-elaboration (pathophysiology added) |
| Murmur: No, Outcome: Abnormal | Provide differential diagnosis. | Possible *valvular dysfunction*, myocardial disease, or congenital anomaly. | Contradiction (valvular dysfunction typically causes murmurs) |

## B.3    Task-Specific Reliability

Table 6 summarizes validation results. Low-entropy tasks (classification, detection, comparison) show 80-90% accuracy, while generative tasks (reasoning, differential diagnosis) show 60-70% accuracy, with errors primarily from clinical over-elaboration.

## B.4    Error Pattern Analysis

**Acoustic Hallucination (2/70, 3%).**    The most concerning error involves adding acoustic features not in annotations. For example, annotations specifying only "wheezing" might generate "wheezing and rhonchi."

Table 6: **Synthetic data validation results by task type.** Based on 10 samples per task (70 total).

| Task | Accurate | Acoustic Halluc. | Clinical Over-elab. | Contradict. |
|---|---|---|---|---|
| Classification | 8/10 (80%) | 0 | 2 | 0 |
| Detection | 9/10 (90%) | 0 | 1 | 0 |
| Reporting | 8/10 (80%) | 1 | 1 | 0 |
| Reasoning | 7/10 (70%) | 0 | 2 | 1 |
| DDx | 6/10 (60%) | 1 | 3 | 0 |
| Comparison | 9/10 (90%) | 0 | 1 | 0 |
| Location | 9/10 (90%) | 0 | 1 | 0 |
| **Total** | **56/70 (80%)** | **2 (3%)** | **11 (16%)** | **1 (1%)** |

Importantly, hallucinated sounds are clinically related to true sounds (both obstructive airway indicators), suggesting the model conflates co-occurring pathological sounds rather than generating arbitrary features.

**Clinical Over-Elaboration (11/70, 16%).** The most frequent error adds clinically plausible but unsupported interpretations. Given only "crackles, pneumonia," the model might add "consistent with fluid accumulation and alveolar collapse." While medically accurate, these explanations extend beyond labels. This is most prevalent in differential diagnosis (3/10) and reasoning (2/10) tasks where instructions explicitly request clinical explanations.

**Contradictions (1/70, 1%).** Rare instances include logical inconsistencies (suggesting "valvular dysfunction" despite "no murmur" annotation). This represent occasional failures in logical consistency.

### B.5 Implications for Benchmark Reliability

Our validation reveals task-dependent reliability: low-entropy tasks achieve 80-90% accuracy, while high-entropy tasks show 60-70% accuracy. Acoustic hallucination (3%) is rare but concerning. Clinical over-elaboration (16%) is more common but less problematic. It reflects medically appropriate interpretations that, while unsupported by minimal labels, demonstrate sound medical reasoning. Contradictions (1%) are rare.

## C  Clinical Accuracy Judgement prompt

Below is the complete prompts for clinical correctness evaluation (See Section 6.1 for details).

---

**LLM prompt for clinical correctness evaluation.**

For the given ground-truth and prediction, determine if the prediction reasonably captures the key information.
Minor stylistic differences, or small missing details should NOT make it a 'No'.
Respond ONLY in JSON with 'answer' as 'Yes' or 'No'. Do NOT provide any explanations.

```
Ground-truth: {row.get('response', '')}
Prediction: {row.get('generated', '')}
```

---

# D  Clinical Accuracy Robustness Analysis

To add robustness and transparency to the clinical accuracy metric (section 6.1), we evaluated 200 randomly sampled instances per task under two conditions: (1) varying judge models with fixed prompt, and (2) varying prompts with fixed judge (GPT-5-mini).

## D.1  Prompt Variations

We designed three prompts with varying wording. **Prompt 1** (baseline, shown in Appendix C) instructs the judge to focus on key information while ignoring minor stylistic differences. **Prompt 2** emphasizing clinical intent. **Prompt 3** applies stricter criteria, requiring that predictions do NOT introduce unsupported clinical claims.

---

**Prompt 2: Lenient (Clinical Intent Focus).**

You are evaluating whether a model output is clinically reasonable.
Given a ground-truth response and a prediction, decide whether the prediction captures the main clinical intent of the ground truth.
Answer "Yes" if:

- The main diagnosis or conclusion is correct, AND

- Any missing details would not meaningfully change clinical interpretation.

Ignore stylistic differences, phrasing, or minor omissions. Do NOT penalize lack of detail unless it changes the clinical meaning.
Respond ONLY in JSON with the key "answer" set to "Yes" or "No". Do NOT provide any explanations.

```
Ground-truth: {row.get('response', '')}
Prediction: {row.get('generated', '')}
```

---

**Prompt 3: Strict (Clinical Consistency Focus).**

You are evaluating a medical AI system.
Given a ground-truth clinical response and a model prediction, determine whether the prediction is clinically consistent with the ground truth.
Answer "Yes" ONLY if:

- The core diagnosis, findings, or clinical conclusions are correct, AND

- The prediction does NOT introduce unsupported clinical claims, overconfident diagnoses, or incorrect reasoning.

Minor stylistic differences are acceptable. Missing or incorrect clinical information SHOULD result in "No".
Respond ONLY in JSON with the key "answer" set to "Yes" or "No". Do NOT provide any explanations.

```
Ground-truth: {row.get('response', '')}
Prediction: {row.get('generated', '')}
```

---

## D.2  Judge Model Sensitivity

Table 7 (left) shows systematic variation across judges. Gemini-2.5-Pro (Comanici et al., 2025) produces consistently higher scores, indicating a more permissive evaluation style, while GPT-5-mini (OpenAI, 2025) and Qwen-Plus (Yang et al., 2025) align closely (differences within 0–6 points). Despite absolute differences,

Table 7: **Robustness analysis of clinical accuracy metric.** Values in parentheses show deviation from the baseline (GPT-5-mini with Prompt 1). All results on 200 random samples per task.

| Task | Judge Model Variation | | | Prompt Variation | | |
|---|---|---|---|---|---|---|
| | GPT-5-mini | Qwen-Plus | Gemini-2.5-Pro | Prompt 1 | Prompt 2 | Prompt 3 |
| Classification | 69.5 | 73.5 (+4.0) | 81.0 (+11.5) | 69.5 | 64.0 (-5.5) | 66.0 (-3.5) |
| Detection | 46.0 | 46.0 (0.0) | 54.5 (+8.5) | 46.0 | 44.5 (-1.5) | 44.5 (-1.5) |
| Report | 32.0 | 34.0 (+2.0) | 56.5 (+24.5) | 32.0 | 30.5 (-1.5) | 30.0 (-2.0) |
| Reasoning | 45.0 | 51.0 (+6.0) | 64.5 (+19.5) | 45.0 | 45.0 (0.0) | 46.0 (+1.0) |
| DDx | 29.0 | 35.0 (+6.0) | 53.0 (+24.0) | 29.0 | 27.5 (-1.5) | 26.0 (-3.0) |
| Comparison | 39.5 | 41.0 (+1.5) | 52.0 (+12.5) | 39.5 | 36.0 (-3.5) | 35.0 (-4.5) |
| Location | 41.5 | 44.0 (+2.5) | 59.5 (+18.0) | 41.5 | 37.5 (-4.0) | 39.5 (-2.0) |

relative task ordering remains consistent: classification achieves highest accuracy, while differential diagnosis remains most challenging.

### D.3 Prompt Sensitivity

Table 7 (right) reveals modest prompt sensitivity. Across all tasks, deviations from the baseline are small (typically within ±5 points), with task rankings stable: classification highest (64.0–69.5%), differential diagnosis lowest (26.0–29.0%).

# E   Additional Baseline Comparisons

Table 8: **Additional baseline comparison.** Performance on in-domain and out-of-domain test sets. BertS denotes BERTScore and Acc denotes Accuracy (both in %). The best score is **bold**, second-best is underlined.

| Model | In-Domain | | Out-of-Domain | |
|---|---|---|---|---|
| | BertS | Acc | BertS | Acc |
| Qwen2.5-Omni | 56.5 | 21.2 | 58.2 | 21.1 |
| Qwen3-Omni | 56.1 | 23.9 | 57.0 | 20.4 |
| Audio Flamingo 3 | 56.2 | 21.4 | 56.7 | 18.3 |
| **StethoLM** | **71.8** | **47.8** | **64.8** | **25.2** |

To strengthen our baseline comparisons (Table 2, 3), we evaluated StethoLM against two additional recent audio-language models: Qwen3-Omni (Xu et al., 2025b) and Audio Flamingo 3 (Goel et al., 2025).

**Models.**

- **Qwen3-Omni**: A 30B parameter Mixture-of-Experts audio-language model trained on 20 million hours of audio data, supporting multimodal understanding and native real-time speech interaction (Xu et al., 2025b).

- **Audio Flamingo 3**: A fully open, state-of-the-art audio–language model that unifies speech, sound, and music understanding with long-audio, multi-turn, and reasoning capabilities.

**Results.**   Table 8 shows that StethoLM consistently outperforms both additional baselines.

**Analysis.**   Both Qwen3-Omni and Audio Flamingo 3 achieve performance comparable to Qwen2.5-Omni, with BERTScores in the 56-57% range and accuracy around 21-24% on in-domain test set. The similar performance levels across these diverse architectures (sequence-to-sequence, cross-attention based, and prefix-tuning approaches) further indicates that general-purpose audio pretraining does not capture the fine-grained acoustic patterns critical for clinical auscultation.

# F  Supplementary Evaluation Metrics

## F.1  ROUGE and METEOR

This section provides supplementary evaluation metrics (ROUGE-1 and METEOR) complementing the primary results in Section 6.3 and Section 6.4. These metrics offer alternative perspectives on generation quality through n-gram overlap (ROUGE-1) and synonym-aware matching (METEOR).

Table 9: Performance comparison on StethoBench benchmark using ROUGE-1 and METEOR (complementing Table 2). R-1 denotes ROUGE-1 and MET denotes METEOR (both in %). The best score for each metric per task is highlighted in **bold**, and the second-best score is underlined. *Note:* "Gemini" refers to **Gemini-2.5-Flash** and "Qwen" refers to **Qwen2.5-Omni**.

| Task | Models | | | | | | | | | | | | | | | |
|---|---|---|---|---|---|---|---|---|---|---|---|---|---|---|---|---|
| | LLM | | Pengi | | LTU | | Gama | | Gemma3N | | Gemini | | Qwen | | StethoLM | |
| | R-1 | MET | R-1 | MET | R-1 | MET | R-1 | MET | R-1 | MET | R-1 | MET | R-1 | MET | R-1 | MET |
| Classification | 20.8 | 15.1 | 4.0 | 0.8 | 31.0 | 20.7 | 18.0 | 12.1 | 29.7 | 19.5 | 14.4 | 16.4 | 33.8 | 22.5 | **50.2** | **36.6** |
| Detection | 13.9 | 9.1 | 2.6 | 0.8 | 19.2 | 10.2 | 15.0 | 9.3 | 22.7 | 15.7 | 12.7 | 14.7 | 24.7 | 13.9 | **43.4** | **32.3** |
| Reporting | 15.0 | 9.4 | 3.6 | 0.9 | 18.0 | 9.8 | 14.8 | 8.2 | 20.3 | 11.5 | 19.0 | 15.4 | 25.6 | 14.3 | **50.0** | **39.3** |
| Reasoning | 13.9 | 6.2 | 1.7 | 0.5 | 18.7 | 7.3 | 19.6 | 9.5 | 18.6 | 8.6 | 10.3 | 14.2 | 26.7 | 14.1 | **48.8** | **35.1** |
| DDx | 14.3 | 8.0 | 1.2 | 0.3 | 18.3 | 10.8 | 14.0 | 7.6 | 14.2 | 7.7 | 6.9 | 10.0 | 25.3 | 15.5 | **35.4** | **24.2** |
| Comparison | 17.6 | 11.8 | 1.3 | 0.7 | 23.7 | 11.0 | 13.4 | 7.2 | 20.1 | 12.2 | 11.5 | 16.7 | 26.8 | 16.0 | **54.5** | **40.7** |
| Location | 20.0 | 11.2 | 0.5 | 0.1 | 26.0 | 13.3 | 19.9 | 10.6 | 25.9 | 14.1 | 12.6 | 16.8 | 27.1 | 15.0 | **44.4** | **33.4** |
| **Overall** | 16.2 | 10.3 | 2.6 | 0.7 | 21.9 | 12.3 | 16.1 | 9.5 | 22.2 | 13.7 | 13.0 | 14.8 | 27.4 | 16.2 | **46.5** | **34.4** |

Table 10: Out-of-Domain (OOD) performance using ROUGE-1 and METEOR (complementing Table 3). R-1 denotes ROUGE-1 and MET denotes METEOR (both in %). The best score for each metric per dataset is highlighted in **bold**, and the second-best score is underlined.

| Dataset | Models | | | | | | | | | | | | | | | |
|---|---|---|---|---|---|---|---|---|---|---|---|---|---|---|---|---|
| | LLM | | Pengi | | LTU | | Gama | | Gemma3N | | Gemini | | Qwen | | StethoLM | |
| | R-1 | MET | R-1 | MET | R-1 | MET | R-1 | MET | R-1 | MET | R-1 | MET | R-1 | MET | R-1 | MET |
| TR | 17.7 | 10.5 | 1.2 | 0.5 | 24.2 | 13.8 | 17.7 | 10.5 | 24.1 | 15.4 | 11.0 | 14.6 | 30.2 | 20.3 | **35.2** | **21.5** |
| CinC | 13.8 | 6.8 | 1.1 | 0.4 | 19.5 | 8.4 | 16.5 | 8.6 | 20.5 | 9.8 | 5.0 | 5.1 | 24.0 | 12.8 | **32.6** | **18.0** |
| BMD | 18.0 | 11.3 | 1.7 | 0.6 | 22.7 | 14.6 | 19.2 | 10.7 | 23.6 | 13.3 | 12.4 | 16.1 | 27.5 | 16.0 | **37.4** | **25.9** |
| FluSense | 16.3 | 12.5 | 2.5 | 0.6 | 23.8 | 12.9 | 19.2 | 12.2 | 22.8 | 16.2 | 15.8 | 14.0 | **30.5** | **18.4** | 25.4 | 17.6 |
| **Overall** | 16.8 | 10.2 | 1.5 | 0.5 | 22.3 | 12.7 | 18.2 | 10.5 | 22.8 | 13.2 | 11.5 | 14.5 | 27.5 | 16.3 | **32.7** | **20.8** |

## F.2  Relaxed Evaluation for Differential Diagnosis

Differential diagnosis is inherently subjective, as it involves ranking conditions by likelihood, a process where expert agreement varies substantially (Elmore et al., 2016). To account for this, we use a relaxed prompt that focuses on clinical plausibility rather than exact ranking match. Using this criterion, StethoLM achieves 35.5% accuracy on DDx tasks on test set (compared to 30.6% before). The relaxed prompt is shown below:

---

**LLM prompt for differential diagnosis (relaxed plausibility).**

For the given differential diagnosis task, assess whether the generated response is clinically appropriate.

A response is clinically appropriate if:

```
1. It includes reasonable differential diagnoses that could explain the findings
2. It does not include clearly inappropriate or unrelated conditions
```

Respond ONLY in JSON with 'answer' as 'Yes' or 'No'. No explanations.

```
Ground-truth: {row.get('response', '')}
Prediction: {row.get('generated', '')}
```

---

# G   Failure Case Analysis

| Inference Query | Response | Failure Analysis |
|---|---|---|
| What is the diagnosis and explanation for the auscultation findings? | ✓ Diagnosis: Tricuspid regurgitation. Holosystolic, harsh murmur at tricuspid area. 
 ✗ Ventricular septal defect, characterized by harsh holosystolic murmur. | **Clinically Similar Acoustic Patterns:** Both conditions produce holosystolic murmurs with similar timing and quality. Differentiation requires radiation patterns, This highlights the value of integrating auscultation with other cardiac exam findings. |
| Describe the type of sound heard. | ✓ Crackles and wet rales indicative of mucus in airways. 
 ✗ Crackles and a dry, non-productive cough are present. | **Fine-Grained Acoustic Discrimination:** Wet vs. dry distinction requires subtle amplitude and spectral cues. Model correctly identifies crackles but misses fluid presence. This suggests current audio encoding may under-emphasize time-frequency characteristics distinguishing moisture content |
| Can you determine any correlation between these sounds and Covid-19? | ✓ Audio 2 correlates with Covid-19 (positive test), Audio 1 is negative. 
 ✗ Audio 1 is associated with positive Covid-19 test, Audio 2 negative. | **Systematic Comparison Reversal:** Complete test status reversal indicates architectural limitation in comparison tasks. Suggests need for explicit audio indexing mechanisms in multi-input scenarios. |
| Give diagnosis and explain the reasoning. | ✓ Diagnosis: Mild COVID-19. Dry cough is typical. 
 ✗ Upper infection. Dry cough and mild severity support this. | **Non-Specific Symptom Challenge:** Dry cough occurs in COVID-19, common cold, allergies, and many other conditions, It lacks distinctive acoustic markers. This error pattern suggests the model hasn't learned reliable COVID-specific acoustic signatures. |
| What acoustic differences can you identify between the two coughs? | ✓ Audio 2 shows a wet, productive cough with crackles, while Audio 1 is dry. 
 ✗ Audio 1 has a wet, productive sound, while Audio 2 is dry. | **Multi-Audio Tracking Limitation:** Model correctly identifies wet/dry characteristics but reverses audio attribution. This architectural challenge, i.e. maintaining audio-label binding across sequential inputs, may be addressed through explicit position encoding or separate encoder pathways. |
| What is the primary sound event detected in this respiratory recording? | ✓ The primary sound event is a sniffle. 
 ✗ The primary sound event is expiratory wheezing. | **Out-of-Protocol Sound Events:** FluSense includes respiratory events (sniffles, gasps, sneezes, throat-clearing) largely absent from clinical auscultation training data, which focuses on pathological sounds (crackles, wheezes, murmurs). |

Figure 6: **Representative failure cases across major error categories.** Each example shows the instruction, ground truth, model prediction, and analysis revealing the underlying failure pattern. These failure patterns complement the successful generation examples in Figure 5, together providing a complete characterization of model capabilities and limitations.

