# OpenReview forum: "StethoLM: Audio Language Model for Cardiopulmonary Analysis Across Clinical Tasks"
_TMLR — Accepted by TMLR_

### Review · Reviewer_EDpX · 2025-12-09

**Summary Of Contributions:**

The paper introduces StethoLM, an audio–language model for cardiopulmonary auscultation, and StethoBench, a large synthetic yet systematically assembled benchmark for instruction-following on medical audio. StethoLM links an EfficientNet audio encoder to a MedGemma-4B-IT backbone and treats auscultation as conditional language modeling, generating reports, explanations, and differential diagnoses rather than single labels. StethoBench aggregates seven public datasets into 16k recordings and 77k instruction–response pairs across seven clinically motivated task families.

Empirically, StethoLM clearly outperforms general audio–language models and large multimodal systems, achieving 71.8% BERTScore and 47.8% accuracy on in-domain data, compared to 56.5% / 21.2% for Qwen-2.5-Omni and near-chance performance for general audio models. Ablations confirm the need for audio input and show benefits from a medical LLM backbone. The model generalizes well to several external clinical datasets and shows competitive zero-shot screening performance, though it struggles on noisy “in-the-wild” audio (FluSense) and on differential diagnosis ranking, and multimodal DPO slightly degrades performance.

I see the main strengths as: (i) a clear formulation of auscultation as instruction-following language modeling; (ii) a substantial and reusable benchmark that unifies diverse datasets into one task space; (iii) thorough comparisons against both audio–language and large multimodal baselines; and (iv) careful analysis of safety and alignment issues, including the observation that multimodal fine-tuning can erode refusal behavior.

The main weaknesses are: (i) limited robustness to unconstrained real-world audio and non-pathological events; (ii) only partial treatment of how to mitigate the safety issues revealed by ablations; and (iii) a still somewhat under-explored story around the failure of DPO and the evaluation of inherently subjective tasks like differential diagnosis.

**Audience:**

Yes

**Audience Explanation:**

This work should interest multiple TMLR audiences. For multimodal and audio–language modeling researchers, it provides a clear case study showing that domain-specific audio and a medical LLM backbone outperform general audio–language and large multimodal models, supported by detailed task-wise results. For medical AI, the shift to instruction-driven clinical reasoning and the introduction of StethoBench offer a solid foundation for developing medical audio agents.

The paper is also relevant to alignment and safety readers: multimodal fine-tuning’s erosion of refusal behavior and the negative multimodal DPO result provide concrete examples of safety regressions and optimization pitfalls. Overall, the benchmark, methods, and safety observations fit well within TMLR’s scope.

**Broader Impact Concerns:**

StethoLM presents several deployment risks. Multimodal fine-tuning causes a loss of refusal, leading the model to give confident diagnoses even without valid audio, which is unsafe for any autonomous clinical use. Its fluent explanations may amplify automation bias, making incorrect outputs more persuasive. Generalization drops on “in-the-wild’’ audio, limiting reliability in home or low-resource settings. Finally, StethoBench’s mixed data sources introduce potential fairness and demographic biases, which warrant caution and future evaluation.

**Claims And Evidence:**

Yes

**Claims Explanation:**

Most central claims are well supported. Table 2 shows clear state-of-the-art performance on StethoBench, with StethoLM substantially outperforming both general audio–language models and large multimodal models across all task types, supporting its positioning as a cardiopulmonary “generalist.” The comparison to general-purpose audio models is strong, as those trained on music or environmental audio transfer poorly, consistent with the need for domain-specific auscultation cues.

Generalization results are balanced: StethoLM is usually best or second-best on external clinical datasets (TR, CinC, BMD), while performing worse on FluSense—a limitation the authors explicitly acknowledge. Ablations further validate design choices, showing audio is essential, medical LLM backbones provide modest gains, and mDPO degrades performance. Zero-shot binary classification results indicate reasonable transfer to downstream screening tasks.

While some clinical implications exceed what offline synthetic-label evaluation can prove, the need for prospective validation is appropriately noted. Overall, the evidence is consistent and proportionate to the claims.

**Requested Changes:**

1. Expand the analysis of safety and hallucination behavior.
The ablation where audio is removed but the multimodal model still produces diagnostic responses is an important safety signal. It would help to unpack this a bit more: for example, whether the projection network or training recipe effectively forces “non-empty” audio features even for silence/null inputs, and whether the refusal degradation appears uniformly across tasks or concentrates in certain instruction types (e.g., explicit “diagnose” prompts). If space permits, a small mitigation experiment, such as adding explicit “null audio → refusal” examples in SFT, or training on corrupted audio with refusal targets would make the claim that StethoLM is a useful foundation for safer downstream systems more convincing.

2. Clarify the DPO setup and discuss why it may have failed.
The multimodal DPO experiments are interesting, but currently a bit under-specified. Since the preference dataset is built by ranking sampled responses with BERTScore against the ground truth, it would be good to spell out the size of this dataset, how many samples per prompt were used, and whether any filtering on length/diversity was applied. A short discussion of why BERTScore may be a problematic proxy for “clinical quality” (e.g., favoring generic but semantically close text, ignoring subtle diagnostic errors) and whether an LLM-based judge or task-specific clinical metric might be more appropriate would add value for readers thinking about DPO in medical settings.

3. Provide more quantitative detail on StethoBench quality control.
The paper notes that around 2% of the synthetic instruction–response pairs were manually reviewed. It would be helpful to report at least an approximate error rate observed in that sample, and to categorize common error types (e.g., hallucinated findings, incorrect anatomy, inconsistent severity descriptions). This would give readers a clearer sense of how noisy the synthetic labels are and where future improvements to the benchmark might be most impactful.

4. Improve consistency in baseline naming and metrics.
In the text, the text-only baseline is described as MedGemma-4B-IT, while in Table 2 it appears under a different label (e.g., “Gemma3N”). Aligning the naming so that readers can immediately see that this column corresponds to the MedGemma backbone used in StethoLM would reduce confusion. Relatedly, the discussion later mentions an “overall accuracy of 44.6%” and “24.6% on out-of-domain data”, whereas Tables 2 and 3 report 47.8% and 25.2% for StethoLM. If these numbers correspond to a slightly different evaluation (e.g., clinical accuracy on a subset of tasks), it would be good to clarify the relationship.

5. Refine evaluation of differential diagnosis.
The differential diagnosis task is inherently subjective because it involves ranking plausible conditions, and the paper already notes this. Since the reported top-1 accuracy is only 30.6% while BERTScore is much higher, I would encourage the authors to consider a complementary metric that gives partial credit, such as top-K accuracy (does the reference diagnosis appear anywhere in the list) or a more tailored LLM-judge prompt that rewards clinically reasonable lists even when the exact order differs. Even a small additional analysis in this direction would help readers interpret the current numbers.

---

> ### Author Response · Authors · 2026-01-13
>
> We sincerely thank the reviewer for the thorough and constructive feedback. We have substantially expanded our safety analysis, clarified preference optimization setup, and provided comprehensive quality validation. Below we detail our responses and corresponding revisions.
>
> ### **1. Safety and hallucination behavior under audio removal**
>
> We have conducted detailed investigation of safety behavior, with expanded analysis added to **Section 7.1** and safety implications discussed in the revised ablation section.
>
> **Key findings from expanded analysis:**
>
> We distinguish two conditions: (i) **null audio** (modality entirely omitted), and (ii) **silent audio** (audio channel present but uninformative waveform). Both exhibit hallucination but through different mechanisms:
>
> - **Silent audio:** The audio encoder produces fixed-length prefix embeddings even for silent waveforms. The language model receives non-empty audio representations and consistently generates clinical responses, often explicitly hallucinating evidence ("the audio shows...", "the sound indicates...").
>
> - **Null audio:** Without audio features, the multimodally fine-tuned model still frequently produces diagnostic responses rather than refusing. This reflects training-induced language prior shift, i.e. the model learns to associate instruction types with auscultation responses and implicitly assumes audio availability.
>
> **Task-level analysis:** This hallucination behavior is **not task-specific**, indicating systemic issues rather than prompt-dependent.
>
> **Mitigation attempt:** We experimented with adding ~1k non-medical audio samples paired with explicit refusal targets. While this increased refusal on invalid inputs, it caused substantial **over-refusal on valid medical audio** (≈30% at inference), demonstrating that naive negative supervision significantly harms clinical recall.
>
> **Practical recommendation:** For near-term deployment, we suggest lightweight pre-filtering (silence detection, non-respiratory audio rejection) before model invocation. Principled refusal calibration that maintains high clinical recall while rejecting invalid inputs remains important future work.
>
>
> ### **2. Clarification of DPO/mDPO setup and failure analysis**
>
> We have clarified that we used **multimodal DPO (mDPO)** with complete setup details and comprehensive failure analysis now in **revised Section 5.2 and Section 7.2**.
>
> **Setup details (Section 5.2):**
> - **Method:** mDPO (Yu et al., 2024) for modality-conditional preferences
> - **Dataset size:** 2400 preference pairs from training set
> - **Construction:** K=5 candidates per prompt at temperatures T ∈ [0.7, 1.3], ranked by BERTScore (best-of-K as chosen, worst-of-K as rejected)
> - **Audio degradation:** On-the-fly temporal cropping, frequency masking, spectral perturbation
>
> **Why mDPO failed (Section 7.2):**
>
> The revised section provides detailed analysis:
> 1. **BERTScore as problematic proxy:** Captures semantic similarity but not diagnostic precision. Rewards generically plausible responses that miss critical clinical details (severity qualifiers, anatomical specificity, temporal characteristics). This is the core issue—BERTScore-based ranking optimizes linguistic quality over clinical correctness.
>
> 2. **Limited headroom after strong SFT:** 70k+ instruction pairs provide solid baseline. Relative reweighting offers minimal benefit and may harm audio-grounded patterns when preference signal misaligns with task objectives.
>
> **Future directions:** Expert-annotated preferences, LLM judges focused on diagnostic validity, diagnostically-targeted degradation strategies.
>
>
> ### **3. Quantitative detail on StethoBench quality control**
>
> We have conducted a manual validation with comprehensive error analysis, detailed in **new Appendix F** and summarized in **revised Section 4.2**.
>
> **Validation results (70 samples, 10 per task):**
> - **Overall: 80% clinically appropriate** (56/70)
> - Low-entropy tasks: 80-90% (classification: 80%, detection: 90%, comparison: 90%, location: 90%)
> - High-entropy tasks: 60-70% (reasoning: 70%, DDx: 60%)
>
> **Error taxonomy with observed prevalence:**
> 1. **Acoustic hallucination (3%):** Adding unsupported sound features (e.g., "wheezing" → "wheezing and rhonchi")
> 2. **Clinical over-elaboration (16%):** Adding plausible but unsupported interpretations—most common error, particularly in DDx and reasoning
> 3. **Contradictions (1%):** Logical inconsistencies
>
> **Key insight:** Errors primarily reflect over-interpretation of minimal labels. Direct contradictions is rare (1%).
>
> **Appendix B** provides complete methodology, task-specific reliability table, representative examples with actual generated text, and detailed error pattern discussion.

---

> ### Author Response · Authors · 2026-01-13
>
> ### **4. Baseline naming and metric consistency**
>
> - **Baseline naming:** We apologize for the confusion caused by Gemma3N, Gemma3N [1] is not another name for the LLM baseline, but a multimodal model supports audio. We added details about Gemma3N in the revised **Section 6.2 - baseline models**, and refered to LLM whenever discuss text-only baseline in the main text.
> - **Metric alignment:** The mentioned "44.6%" and "24.6%" was not correctly updated; all tables now report consistent final evaluation results (47.8% in-domain, 25.2% OOD for StethoLM overall accuracy)
>
>
> ### **5. Refine evaluation of differential diagnosis**
>
> We have implemented the reviewer's suggestion of a more tailored LLM-judge prompt for differential diagnosis, with results in **Appendix F.2**.
>
> We developed relaxed evaluation focusing on clinical plausibility rather than strict ordering. The prompt assesses whether generated differentials are clinically appropriate (reasonable diagnoses, no inappropriate conditions), allowing ranking variations.
>
> **Results:** Relaxed DDx accuracy: **35.5%** (vs. 30.6% strict).
>
> **Methodological choice:** We report this as supplementary analysis (Appendix F) rather than replacing main metrics to maintain consistency across all seven task categories.
>
> [1]for Google DeepMind. Gemma 3n e4b. https://huggingface.co/google/gemma-3n-E4B, 2025. Multimodal instruction-tuned model supporting audio, image, video, and text.

---

### Review · Reviewer_VQyZ · 2025-12-12

**Summary Of Contributions:**

This paper introduces StethoLM, an audio language model specialized for cardiopulmonary auscultation that moves beyond narrow classification toward instruction-following clinical reasoning. The authors also present StethoBench, a large benchmark of over 77k instruction response pairs derived from diverse heart and lung sound datasets, covering tasks such as reporting, reasoning, differential diagnosis, comparison, and location-based analysis. The work demonstrates that domain-specific audio language training yields substantial gains over general-purpose multimodal models, including stronger performance on out-of-distribution data. Strengths include the clear motivation, comprehensive benchmark design, strong experimental evaluation, and thoughtful discussion of limitations. A key weakness is reliance on synthetic instruction data and automated clinical evaluation, which raises questions about real-world clinical validity.

**Additional Comments:**

This is a well-written and carefully executed paper that makes a meaningful contribution to medical audio language modeling. The benchmark and task framing are particularly valuable and likely to influence future work in the area. With minor clarifications around evaluation and safety considerations, the work would be a strong fit for TMLR.

**Audience:**

Yes

**Audience Explanation:**

The paper will be of interest to researchers working on multimodal learning, foundation models in healthcare, and audio understanding. Its focus on instruction-following models, benchmark construction, and evaluation challenges in medical AI aligns well with TMLR’s audience, even for readers not directly focused on clinical audio.

**Broader Impact Concerns:**

The paper appropriately frames StethoLM as a clinical decision support tool rather than an autonomous diagnostic system. A notable concern is the model's tendency to generate confident outputs even when audio input is missing, which could pose safety risks in deployment settings. While this issue is acknowledged, a clearer discussion of mitigation strategies, such as refusal mechanisms or uncertainty estimation, would strengthen the broader impact discussion.

**Claims And Evidence:**

Yes

**Claims Explanation:**

The claims are well supported by extensive quantitative and qualitative evaluation. The paper includes comparisons against strong baselines, ablation studies, and both in-domain and out-of-domain testing. The results consistently show that StethoLM benefits from domain-specific audio language training and outperforms general audio language models. The authors are careful to contextualize limitations, particularly for subjective tasks such as differential diagnosis, and do not overclaim clinical readiness.

**Requested Changes:**

1. Clarify the clinical reliability of synthetic instruction response generation by expanding the manual validation process or providing additional analysis of error patterns tied to synthetic data generation. This is important for strengthening confidence in the benchmark.

2. Provide more detail on the LLM-based clinical accuracy evaluation, including potential biases of the judge model and sensitivity to prompt design. This would improve transparency around the reported accuracy metric.

3. Consider adding a small expert-annotated evaluation subset or inter-rater comparison to better ground the results in real clinical practice. This would strengthen the paper, but may not be strictly required for acceptance.

---

> ### Author Response · Authors · 2026-01-13
>
> We sincerely thank the reviewer for the thoughtful comments on clinical reliability and evaluation transparency. We have substantially expanded our validation and robustness analyses in the revised manuscript uploaded. Below we detail our responses and corresponding changes.
>
>
> ### **1. Clinical reliability of synthetic instruction–response generation**
>
> We have conducted systematic manual validation with results detailed in **Appendix B** and summarized in **revised Section 4.2**.
>
> **Main findings:** We validated 70 random samples (10 per task) for factual consistency and clinical plausibility. Overall accuracy is **80%** (56/70 clinically appropriate), with straightforward tasks achieving 80-90% (classification, detection, comparison, location) and generative tasks 60-70% (reasoning: 70%, DDx: 60%).
>
> **Error patterns identified:**
> - Acoustic hallucination (3%): Adding unsupported sound features
> - Clinical over-elaboration (16%): Adding plausible but unsupported interpretations (most common)
> - Contradictions/inappropriate responses (1%): Logical inconsistencies (very rare)
>
> Most errors represent clinically plausible elaborations rather than factual inaccuracies. Direct contradictions are rare (1%).  **Appendix B** provides complete validation methodology, task-specific reliability table, error taxonomy with examples, and detailed discussion.
>
> ### **2. Transparency of LLM-based clinical accuracy evaluation**
>
> We have conducted comprehensive robustness analysis, now detailed in **new Appendix D (Clinical Accuracy Robustness Analysis)** and referenced in revised **Section 6.1**.
>
> **Experimental design (200 random samples per task, 1,400 total):**
> - **Judge model variation:** GPT-5-mini, Qwen-Plus, Gemini-2.5-Pro (fixed Prompt 1)
> - **Prompt variation:** Prompt 1 (default/conservative), Prompt 2, Prompt 3 (fixed GPT-5-mini)
>
> **Key findings (summarized from Appendix table):**
>
> *Judge model sensitivity:* Gemini-2.5-Pro systematically more permissive (+10-25 points depending on task), while GPT-5-mini and Qwen-Plus closely aligned (within 2-6 points). For example, classification: GPT-5-mini 69.5%, Qwen-Plus 73.5%, Gemini 81.0%.
>
> *Prompt sensitivity:* Modest variation across prompt wordings (typically ±2-5 points). For example, classification: Prompt 1: 69.5%, Prompt 2 (lenient): 64.0%, Prompt 3 (strict): 66.0%.
>
> *Consistent task ranking:* Critically, **relative task difficulty rankings remain consistent** across all judge models and prompts—classification consistently highest, DDx consistently lowest.
>
> **Section 6.1 now explicitly mentions:** "To validate robustness and add transparency, we conducted sensitivity analyses across multiple judge models and prompt variations (Appendix D), revealing that while absolute scores vary across judges, task-level difficulty rankings remain consistent."
>
> We believe this comprehensive robustness analysis provides more transparency and demonstrates that our evaluation methodology.
>
>
> ### **3. Expert-annotated evaluation or inter-rater comparison**
>
> We agree that expert-annotated evaluation or inter-rater comparison would further strengthen the clinical grounding of the benchmark. However, conducting a reliable expert study within the rebuttal timeline is not feasible as it requires domain-specific clinician involvement. We explicitly acknowledge the absence of expert adjudication as a limitation and identify it as an important direction for future work in **revised Section 8 - limitations**.

---

### Review · Reviewer_fppt · 2025-12-30

**Summary Of Contributions:**

The paper focuses on cardiopulmonary auscultation beyond classification tasks. It proposes StethoLM, an audio-language model, and StethoBench, a benchmark with 77k instruction-response pairs across seven clinical task categories including binary classification, detection, reporting, reasoning, differential diagnosis, comparison and location-based analysis.

**Audience:**

Yes

**Audience Explanation:**

Both the proposed model, which is simple yet effective, and the benchmark will be of interest to the community. It is important to consider tasks that are aligned with clinical practice for this application and this work will be a step towards that direction.

**Claims And Evidence:**

Yes

**Claims Explanation:**

The paper claims that StethoLM achieves gains in performance on IID and OOD tasks which has been supported by the overall evaluation.

**Requested Changes:**

Although the overall paper is well-written, I believe the following changes are essential to improve the positioning and evaluation of the paper:
- It is currently unclear what does out of distribution mean in the evaluated setting and why are the evaluated datasets out of distribution. Is it a shift in $p(x)$ while keeping $p(y|x)$ constant or $p(y|x)$ shifts? How are the in and out of distribution datasets chosen?
- The claim that audio grounding is essential is not fully supported in the paper. While the paper does evaluate with audio and language baselines and removes audio altogether during inference in ablation, I'd suggest to conduct sanity checks from recent multi-modal papers [1,2]. Removal can often be considered out of distribution by the model and thus shuffling the input modality is a better choice. In addition it is important to consider other associations with the ground truth and label to ensure the necessity of audio modality.
- The baseline models can be strengthened to position the work better. I'd suggest to compare with Qwen3-Omni [3], given the strong performance of Qwen2.5-Omni in the experiments. The paper should also include a comparison with the multi-modal large language models discussed in the related work section.
- The paper should elaborate on why performance optimization does not improve audio language model performance in section 7.2. It'd also be useful to have the ablation comparison between base model, SFT and DPO across multiple tasks to support this discussion.
- The paper should discuss more on the metric choices. For instance, why were BertScore preferred for main paper while Rouge and Meteor where positioned in the appendix? The paper should also elaborate on the discrepancies in BertScore and accuracy for both IID and OOD tasks.
- A minor comment is to follow the same template for Table 2 and Table 3. Either having models as the first row or the first column will make it easier for the readers to follow.

References.
[1] Gu et al. The Illusion of Readiness: Stress Testing Large Frontier Models on Multimodal Medical Benchmarks.
[2] Madaan et al. Multi-modal Data Spectrum: Multi-modal Datasets are Multi-dimensional.
[3] Xu et al. Qwen3-Omni Technical Report

---

> ### Author Response · Authors · 2026-01-13
>
> We sincerely thank the reviewer for the thorough and constructive feedback. We have substantially revised the manuscript (see the updated paper, changed highlighted in blue) to address each concern. Below we detail our responses and the corresponding changes.
>
>
> ### **1. Definition of out-of-distribution (OOD) evaluation**
>
> We have clarified the OOD definition with explicit taxonomy and concrete examples in the revised manuscript.
>
> **What "OOD" means in our setting:** Our OOD evaluation reflects realistic deployment scenarios where models encounter **compound distribution shifts** across multiple dimensions. We now explicitly define three types of shift in **Section 4.1**:
>
> 1. Covariate shift p(x): Different recording devices (TR: 12-channel vs single-point stethoscopes), acoustic environments, patient populations (BMD: adult valvular pathologies vs CirCor/ZCHSound: pediatric-focused training)
> 2. Label shift p(y): Different disease prevalence distributions across populations
> 3. Changed label spaces (FluSense: spontaneous events like sneezes/coughs vs training data: pathological sounds like crackles/wheezes)
>
> **Dataset selection rationale:** In-domain and OOD datasets were chosen based on: (1) no overlap with training sources, (2) distinct clinical contexts, (3) realistic deployment relevance, and (4) public availability with sufficient metadata.
>
>
> ### **2. Evidence for audio grounding and multimodal sanity checks**
>
> Following the reviewer's suggestion and recent multimodal evaluation practices, we conducted **audio-text mismatch experiments** by randomly shuffling audio across samples while keeping instructions unchanged.
>
> **Results (added to Section 7.1, Table 4):**
> - Audio-text shuffled: BERTScore 71.8 → 63.6, Accuracy 47.8 → 31.9
>
> The shuffling experiment demonstrates that the model relies on **semantic audio-text correspondence** rather than exploiting spurious correlations or text-only patterns. We note that because many pathologies appear multiple times in the dataset (e.g., multiple COPD recordings with wheezing), some shuffled pairs remain accidentally label-compatible, making this a conservative underestimate of the true performance degradation.
>
>
> ### **3. Strengthening baseline comparisons**
>
> We have added **Qwen3-Omni** and **Audio Flamingo 3** as additional baselines, with comprehensive results now provided in **Appendix E (Additional Baseline Comparisons)**.
>
> **Summary results:**
>
> | Model            | In-Domain BERTScore | In-Domain Acc | OOD BERTScore | OOD Acc |
> |------------------|--------------------:|--------------:|--------------:|--------:|
> | Qwen2.5-Omni     | 56.5                | 21.2          | 58.2          | 21.1    |
> | Qwen3-Omni       | 56.1                | 23.9          | 57.0          | 20.4    |
> | Audio Flamingo 3 | 56.2                | 21.4          | 56.7          | 18.3    |
> | **StethoLM**     | **71.8**            | **47.8**      | **64.8**      | **25.2**|
>
> While Qwen3-Omni shows modest in-domain accuracy improvements over Qwen2.5 (+2.7 pp), StethoLM maintains substantial advantages across both settings. Regarding other multimodal LLMs from related work: Gemini-2.5-Flash (included in Table 2) represents the state-of-the-art multimodal baseline with audio support, while most other discussed models either lack native audio capabilities or full instruction-following for medical reasoning tasks.
>
>
> ### **4. Why preference optimization (mDPO) does not consistently help**
>
> We have expanded **Section 7.2** with comprehensive failure analysis.
>
> **Key findings:** Despite successful training-time preference learning (78% response preference accuracy, 71% audio preference accuracy), test performance degraded (BERTScore: 71.8% → 69.8%; Accuracy: 47.8% → 45.9%). The revised section now provides detailed analysis of two domain-specific factors:
>
> 1. **BERTScore as problematic preference proxy:** Captures semantic similarity but not diagnostic precision. May reward generically plausible responses that miss critical clinical details (severity qualifiers, anatomical specificity, temporal characteristics). This causes preference optimization to refine linguistic surface patterns rather than diagnostic reasoning.
>
> 2. **Limited headroom after strong SFT:** 70k+ instruction-response pairs provide solid baseline supervision. Relative preference reweighting offers minimal benefit and may harm audio-grounded patterns when the preference signal misaligns with task objectives.

---

> > ### Comment · Reviewer_fppt · 2026-01-30
> > **Response to rebuttal**
> >
> > Thank you for your rebuttal, please find my responses below:
> >
> > 1. **Definition of OOD distribution:** Thank you including additional discussion in section 4.1. Can you further elaborate on how the metrics are reported? Do you take an average over all the types? It would be useful to include both average and worse performance and include details of individual shifts.
> > 2. **Sanity checks:** Unfortunately, the shuffling by itself does not inform us about the significance of those numbers. It is important to include a random baseline and shuffle the text as well. The same holds for zeroing out as well. This is essential to understand how much benefit does audio really provide.
> > 3. **Additional baselines:** It is interesting to observe that all the additional baselines perform worse than the earlier considered baselines. Thank you for running these experiments.
> > 4. **mDPO:** Regarding BERTScore, did you see contradicting results for other metrics as well? If yes, it is unclear if it is a metric issue. Otherwise, it would be useful to include a discussion on the differences.

---

> > > ### Author Response · Authors · 2026-01-31
> > > **Response to follow-up**
> > >
> > > We thank the reviewer for the follow up. In the following we address each point.
> > >
> > > **1.OOD distribution**
> > >
> > > To clarify our current reporting, Table 3 (OOD performance) presents individual performance for each OOD dataset alongside the overall OOD average. Regarding shift-type-specific analysis, we appreciate this suggestion but conducting such analysis is not feasible at this time due to (1) the limited publicly available respiratory and cardiac audio datasets we have, and (2) each dataset exhibits compound shifts rather than isolated shift types. We agree that shift-type-specific analysis would be valuable future work and would take it into consideration when sourcing ood datasets.
> > >
> > >
> > > **2.Sanity checks**
> > >
> > > To fully address the concern and provide a holistic comparison, we ran additional ablations around modality contribution. Results on the test set
> > >
> > > | Condition | BERTScore | Accuracy | Interpretation |
> > > |-----------|-----------|----------|----------------|
> > > | **Both shuffled** | 49.4 | 5.7 | Random pairing - near-floor performance |
> > > | **Text shuffled** | 51.9 | 10.2 | Wrong task instruction for given audio |
> > > | **Audio shuffled** | 63.6 | 31.9 | Wrong audio for given task |
> > > | **Audio removed** | 57.8 | 28.5 | Instruction-only, no audio signal |
> > > | **StethoLM (both correct)** | **71.8** | **47.8** | Full performance |
> > >
> > > We want to highlight that the model is trained to follow instructions—i.e., to **perform specific tasks conditioned on both instruction and audio**. Shuffling audio creates audio mismatch while preserving valid instruction-task pairing, directly testing whether the model exploits audio content vs. text patterns. This is why we choose to provide audio-shuffling in the previous rebuttal. In contrast, **shuffling text creates a fundamentally different kind of mismatch**: the instruction may ask about respiratory conditions while the paired audio and ground truth concern heart murmur detection. This cross-task misalignment induces larger degradation (10.2% vs. 31.9%) but is expected.
> > >
> > > We hope our interpretation of the reviewer’s comments is accurate, and we appreciate the reviewer’s insightful guidance on relevant ablation studies.
> > >
> > >
> > >
> > > **3.Additional baseline**
> > >
> > > Actually Qwen3-Omni achieves the highest in-domain accuracy, though most recent SOTA models perform similarly. This indicates that the bottleneck is medical domain-specificity. Even with advanced reasoning, these general models fail to recognize the **specific acoustic patterns tied to pathology**.
> > >
> > > **4.(m)DPO**
> > >
> > > We would like to clarify whether the reviewer is asking if we explored alternative metrics as preference signals, or if we observed performance decreases across metrics other than BERTScore.
> > >
> > > We only explored BERTScore as the preference signal due to its simplicity, low cost, and widespread use in generation tasks. Given sufficient resources, clinician-derived preferences (capturing diagnostic prioritization and clinical communication quality) would represent a more promising direction. However, the core challenge may extend beyond metric choice: DPO is most effective when tasks are subjective with no single correct answer. Medical audio interpretation has relatively objective correctness criteria, and the strong SFT baseline (70k pairs) may leave limited headroom for BERTScore-based preference optimization.

---

> ### Author Response · Authors · 2026-01-13
>
> ### **5. Metric choices**
>
> We have added explicit metric justification in **Section 6.1**.
>
> **Why BERTScore:** We now explicitly explain three reasons: (1) contextualized embeddings (DeBERTa-large-mnli) capture semantic equivalence in medical language better than n-gram metrics (e.g., recognizing "basilar crepitations" and "crackles at lung bases" as semantically equivalent), (2) alignment with recent medical multimodal model evaluation practices (MedGemma, MedRAX), and (3) continuous measurement of partial correctness. ROUGE and METEOR provide complementary lexical perspectives and remain in Appendix.
>
> We emphasize that reporting both metrics is deliberate: BERTScore measures linguistic/semantic alignment while LLM-judged accuracy evaluates diagnostic validity. The divergence provides insight into model capabilities and failure modes.
>
>
> ### **6. Table formatting**
>
> We have revised **Table 3** to use models-as-columns format matching Table 2, improving readability and consistency; as well as table 9 in Appendix - Supplementary Evaluation Metrics.

---

### Decision · Action_Editor_2iUD · 2026-02-07

**Recommendation:** Accept as is

**Audience:**

Yes

**Audience Explanation:**

Though the approach, fine-tuning a model for a specific domain, is known, the application is interesting. The paper misses the opportunity to discuss the difficulty of analyzing cardiopulmonary recordings and whether the generated reports are useful for clinical experts. However, these are also the reasons that we need more people to work on this problem. The paper is easy to follow and reduces the barrier for others to work on this problem.

**Claims And Evidence:**

Yes

**Claims Explanation:**

This is a system-building type of paper, where MedGemma-4B-IT is fine-tuned for analyzing cardiopulmonary sounds and answering queries about cardiopulmonary auscultation. To evaluate the system, the paper aggregates several tasks and repurpose datasets for answering queries about cardiopulmonary auscultation.

There aren't many claims throughout the paper, except maybe the claim that there are limited studies on audio Q&A in the clinical domain. This paper indeed fills in this gap.

All reviewers also agree that the paper is sound and the contribution is significant.